# communications
# engineering

# Long term 5G network traffic forecasting via modeling non-stationarity with deep learning

Yuguang Yang[1,5], Shupeng Geng [1,5], Baochang Zhang [1,2✉], Juan Zhang [1,2✉], Zheng Wang[3], Yong Zhang[3] & David Doermann[4]

5G cellular networks have recently fostered a wide range of emerging applications, but their popularity has led to traffic growth that far outpaces network expansion. This mismatch may decrease network quality and cause severe performance problems. To reduce the risk, operators need long term traffic prediction to perform network expansion schemes months ahead. However, long term prediction horizon exposes the non-stationarity of series data, which deteriorates the performance of existing approaches. We deal with this problem by developing a deep learning model, Diviner, that incorporates stationary processes into a well-designed hierarchical structure and models non-stationary time series with multi-scale stable features. We demonstrate substantial performance improvement of Diviner over the current state of the art in 5G network traffic forecasting with detailed months-level forecasting for massive ports with complex flow patterns. Extensive experiments further present its applicability to various predictive scenarios without any modification, showing potential to address broader engineering problems.

[1] Beihang University, 100191 Beijing, China. [2] Zhongguancun Laboratory, 100094 Beijinig, China. [3] China Unicom, 100037 Beijing, China. [4] University at Buffalo, 14260 Buffalo, NY, USA. [5] These authors contributed equally: Yuguang Yang and Shupeng Geng. ✉email: bczhang@buaa.edu.cn; zhang_juan@buaa.edu.cn

**5G** technology has recently gained popularity worldwide for its faster transfer speed, broader bandwidth, reliability, and security. 5G technology can achieve a 20× faster theoretical peak speed over 4G with lower latency, promoting applications like online gaming, HD streaming services, and video conferences[1–3]. The development of 5G is changing the world at an incredible pace and fostering emerging industries such as telemedicine, autonomous driving, and extended reality[4–6]. These and other industries are estimated to bring a 1000-fold boost in network traffic, requiring the additional capacity to accommodate these growing services and applications[7]. Nevertheless, 5G infrastructure, such as board cards and routers, must be deployed and managed with strict cost considerations[8,9]. Therefore, operators often adopt a distributed architecture to avoid massive back-to-back devices and links among fragmented networks[10–13]. As shown in Fig. 1a, the emerging metropolitan router is the hub to link urban access routers, where services can be accessed and integrated effectively. However, the construction cycle of 5G devices requires about three months to schedule, procure, and deploy. Planning new infrastructures requires accurate network traffic forecasts months ahead to anticipate the moment that capacity utilization surpasses the preset threshold, where the overloaded capacity utilization might ultimately lead to performance problems. Another issue concerns the resource excess caused by building coarse-grained 5G infrastructures. To mitigate these hazards, operators formulate network expansion schemes months ahead with long-term network traffic prediction, which can facilitate long-period

planning for upgrading and scaling the network infrastructure and prepare it for the next planning period[14–17].

In industry, a common practice is calculating network traffic's potential growth rate by analyzing the historical traffic data[18]. However, this approach cannot scale to predict the demand for new services and is less than satisfactory for long-term forecasting. And predictions-based methods have been introduced to solve this dilemma by exploring the potential dependencies involved in historical network traffic, which provides both a constraint and a source for assessing future traffic volume. Network planners can harness the dependencies to extrapolate long-enough traffic forecasts to develop sustainable expansion schemes and mitigation strategies. The key issue to this task is to obtain an accurate long-term network traffic prediction. However, directly extending the prediction horizon of existing methods is ineffective for long-term forecasting since these methods suffer a severe performance degeneration, where the long-term prediction horizon exposes the non-stationarity of time series. This inherent non-stationarity of real-world time series data is caused by multi-scale temporal variations, random perturbations, and outliers, which present various challenges. These are summarized as follows. (a) Multi-scale temporal variations. Multi-scale (daily/weekly/monthly/yearly) variations throughout long-term time series indicate multi-scale non-stationary latent patterns within the time series, which should be taken into account comprehensively in the model design. The seasonal component, for example, merely presents variations at particular scales. (b) Random factors. Random perturbations and outliers interfere

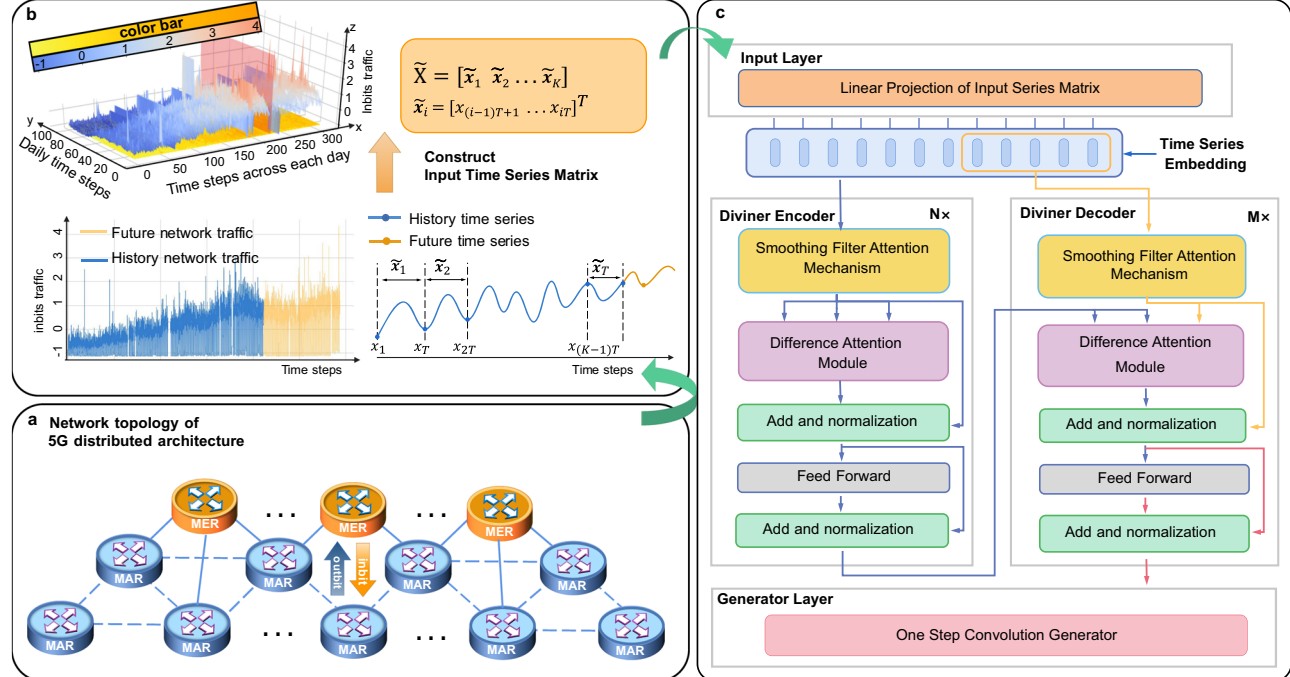

**Fig. 1 Schematic illustration for the workflow of Diviner. a** We collect the data from MAR–MER links. The orange cylinder depicts the metropolitan emerging routers (MER), and the pale blue cylinder depicts metropolitan accessing routers (MAR). **b** The illustration of the introduced 2D → 3D transformation process. Specifically, given a time series of network traffic data spanning $K$ days, we construct a time series matrix $\widetilde{\mathbf{X}} = [\tilde{\mathbf{x}}_1 \; \tilde{\mathbf{x}}_2 \; \ldots \; \tilde{\mathbf{x}}_K]$, where each $\tilde{\mathbf{x}}_i$ represents the traffic data for a single day of length $T$. The resulting 3D plot displays time steps across each day, daily time steps, and inbits traffic along the $x$, $y$, and $z$ axes, respectively, with the inbits traffic standardized. The blue line in the 2D plot and the side near the origin of the pale red plane in the 3D plot represent historical network traffic, while the yellowish line in the 2D plot and the side far from the origin of the pale red plane in the 3D plot represent the future network traffic to predict. **c** The overall working flow of the proposed Diviner. The blue solid line indicates the data stream direction. Both the encoder and decoder blocks of Diviner contain a smoothing filter attention mechanism (yellowish block), a difference attention module (pale purple block), a residual structure (pale green block), and a feed-forward layer (gray block). Finally, a one-step convolution generator (magenta block) is employed to convert the dynamic decoding into a sequence-generating procedure.

with the discovery of stable regularities, which entails higher robustness in prediction models. (c) Data distribution shift. Non-stationarity of the time series inevitably results in a dataset shift problem with the input data distribution varying over time. Figure 1b illustrates these challenges.

Next, we review the shortcomings of existing methods concerning addressing non-stationarity issues. Existing time series prediction methods generally fall into two categories, conventional models and deep learning models. Most conventional models, such as ARIMA[19,20] and HoltWinters[21–25], are built with some insight into the time series but implemented linearly, causing problems for modeling non-stationary time series. Furthermore, these models rely on manually tuned parameters to fit the time series, which impedes their application in large-scale prediction scenarios. Although Prophet[26] uses a nonlinear modular and interpretive parameter to address these problems, its hand-crafted nonlinear modules need help to easily model non-stationary time series, whose complex patterns make it inefficient to embed diverse factors in hand-crafted functions. This dilemma boosts the development of deep learning methods. Deep learning models can utilize multiple layers to represent latent features at a higher and more abstract level[27], enabling us to recognize deep latent patterns in non-stationary time series. Recurrent neural networks (RNNs) and Transformer networks are two main deep learning forecasting frameworks. RNN-based models[28–34] feature a feedback loop that allows models to memorize historical data and process variable-length sequences as inputs and outputs, which calculates the cumulative dependency between time steps. Nevertheless, such indirect modeling of temporal dependencies can not disentangle information from different scales within historical data and thus fails to capture multi-scale variations within non-stationary time series. Transformer-based models[35–37] solve this problem using a global self-attention mechanism rather than feedback loops. Doing so enhances the network's ability to capture longer dependencies and interactions within series data and thus brings exciting progress in various time series applications[38]. For more efficient long-term time series processing, some studies[39–41] turn the self-attention mechanism into a sparse version. However, despite their promising long-term forecasting results, time series' specialization is not taken into account during their modeling process, where varying distributions of non-stationary time series deteriorate their predictive performances. Recent research attempts to incorporate time series decomposition into deep learning models[42–47]. Although their results are encouraging and bring more interpretive and reasonable predictions, their limited decomposition, e.g., trend-seasonal decomposition, reverses the correlation between components and merely presents the variation of time series at particular scales.

In this work, we incorporate deep stationary processes into neural networks to achieve precise long-term 5G network traffic forecasts, where stochastic process theories can guarantee the prediction of stationary events[48–50]. Specifically, as shown in Fig. 1c, we develop a deep learning model, Diviner, that incorporates stationary processes into a well-designed hierarchical structure and models non-stationary time series with multi-scale stable features. To validate the effectiveness, we collect an extensive network port traffic dataset (NPT) from the intelligent metropolitan network delivering 5G services of China Unicom and compare the proposed model with numerous current arts over multiple applications. We make two distinct research contributions to time series forecasting: (1) We explore an avenue to solve the challenges presented in long-term time series prediction by modeling non-stationarity in the deep learning paradigm. This line is much more universal and effective than the previous works incorporating temporal decomposition for their limited decomposition that merely presents the temporal variation at particular scales. (2) We develop a deep learning framework with a well-designed hierarchical structure to model the multi-scale stable regularities within non-stationary time series. In contrast to previous methods employing various modules in the same layer, we perform a dynamical scale transformation between different layers and model stable temporal dependencies in the corresponding layer. This hierarchical deep stationary process synchronizes with the cascading feature embedding of deep neural networks, which enables us to capture complex regularities contained in the long-term histories and achieve precise long-term network traffic forecasting. Our experiment demonstrates that the robustness and predictive accuracy significantly improve as we consider more factors concerning non-stationarity, which provides an avenue to improve the long-term forecast ability of deep learning methods. Besides, we also show that the modeling of non-stationarity can help discover nonlinear latent regularities within network traffic and achieve a quality long-term 5G network traffic forecast for up to three months. Furthermore, we expand our solution to climate, control, electricity, economic, energy, and transportation fields, which shows the applicability of this solution to multiple predictive scenarios, showing valuable potential to solve broader engineering problems.

## Results

**Diviner with deep stationary processes**. In this Section, we introduce our proposed deep learning model, Diviner, that tackles the non-stationarity of long-term time series prediction with deep stationary processes, which captures multi-scale stable features and models multi-scale stable regularities to achieve long-term time series prediction.

*Smoothing filter attention mechanism as a scale converter.* As shown in Fig. 2a, the smoothing filter attention mechanism adjusts the feature scale and enables Diviner to model time series from different scales and access the multi-scale variation features within non-stationary time series. We build this component based on Nadaraya-Watson regression[51,52], a classical algorithm for non-parametric regression. Given the sample space $\Omega = \{(x_i, y_i) | 1 \leq i \leq n, x_i \in \mathbb{R}, y_i \in \mathbb{R}\}$, window size $h$, and kernel function $K(\cdot)$, the Nadaraya–Watson regression has the following expression:

$$\hat{y} = \sum_{i=1}^{n} K\left(\frac{x - x_i}{h}\right) y_i \Big/ \sum_{j=1}^{n} K\left(\frac{x - x_j}{h}\right), \tag{1}$$

where the kernel function $K(\cdot)$ is subject to $\int_{-\infty}^{\infty} K(x)dx = 1$ and $n, x, y$ denote sample size, independent variable, and dependent variable, respectively.

The Nadaraya–Watson regression estimates the regression value $\hat{y}$ using a local weighted average method, where the weight of a sample $(x_i, y_i)$, $K(\frac{x-x_i}{h}) / \sum_{j=1}^{n} K(\frac{x-x_j}{h})$, decays with the distance of $x_i$ from $x$. Consequently, the primary sample $(x_i, y_i)$ is closer to samples in its vicinity. This process implies the basic notion of scale transformation, where adjacent samples get closer on a more significant visual scale. Inspired by this thought, we can reformulate the Nadaraya–Watson regression from the perspective of scale transformation. We incorporate it into the attention structure to design a learnable scale adjustment unit. Concretely, we introduce the smoothing filter attention mechanism with a learnable kernel function and self-masked operation, where the former shrinks (or magnifies) variations for adaptive feature-scale adjustment, and the letter eliminates outliers. To ease understanding, we consider the 1D time series case here, and the high-dimensional case can be easily extrapolated (shown mathematically in Section "Methods"). Given the time step $t_i$, we estimate its

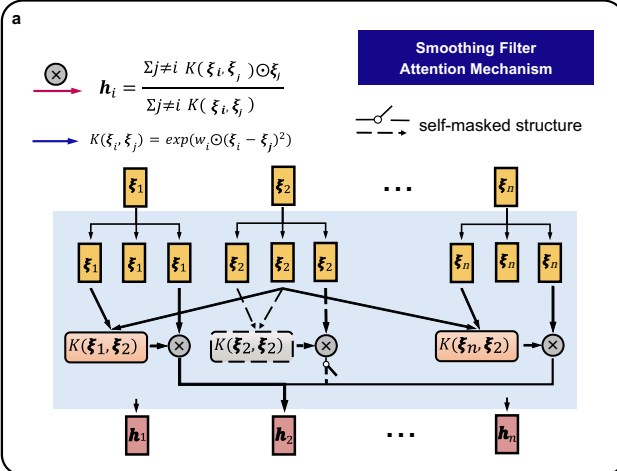

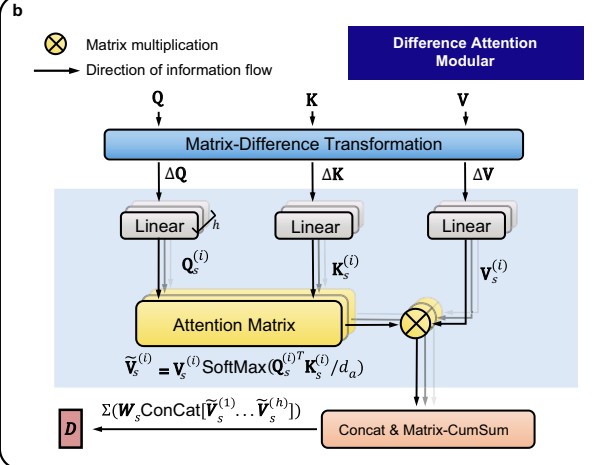

**Fig. 2 Illustration of the structure of smoothing filter attention mechanism and difference attention module. a** This panel displays the smoothing filter attention mechanism, which involves computing adaptive weights $K(\xi_i, \xi_j)$ (orange block) and employing a self-masked structure (gray block with dashed lines) to filter out the outliers, where $\xi_i$ denotes the $i$th embedded time series period (yellow block). The adaptive weights serve to adjust the feature scale of the input series and obtain the scale-transformed period embedding $\mathbf{h}_i$ (pink block). **b** This diagram illustrates the difference attention module. The Matrix-Difference Transformation (pale blue block) subtracts adjacent columns of a matrix to obtain the shifted query, key, and value items ($\Delta\mathbf{Q}$, $\Delta\mathbf{K}$, and $\Delta\mathbf{V}$). Then, an autoregressive multi-head self-attention is performed (in the pale blue background) to capture the correlation of time series across different time steps, resulting in $\widetilde{\mathbf{V}}_s^{(i)}$ for the $i$th attention head. Here, $\mathbf{Q}_s^{(i)}$, $\mathbf{K}_s^{(i)}$, $\mathbf{V}_s^{(i)}$, and $\widetilde{\mathbf{V}}_s^{(i)}$ represent the query, key, value, and result in items, respectively. the SoftMax is applied to the scaled dot-product between the query and key vectors to obtain attention weights (the pale yellow block). The formula for the SoftMax function is $\mathrm{SoftMax}(\mathbf{k}_i) = e^{\mathbf{k}_i}/\sum_{j=1}^n e^{\mathbf{k}_j}$, where $\mathbf{k}_i$ is the $i$th element of the input vector, and $n$ is the length of the input vector. Lastly, the Matrix-CumSum operation (light orange block) accumulates the shifted features using the ConCat operation, and $\mathbf{W}_s$ denotes the learnable aggregation parameters.

regression value $\hat{y}_i$ with an adaptive-weighted average of values $\{y_t | t \neq t_i\}$, $\hat{y}_i = \sum_{j \neq i} \alpha_j y_j$, where the adaptive weights $\boldsymbol{\alpha}$ are obtained by a learnable kernel function $f$. The punctured window $\{t_j | t_j \neq t_i\}$ of size $n-1$ denotes our self-masked operation, and $f(y_i, y)_{w_i} = exp(w_i(y_i - y)^2)$, $\alpha_i = f(y_i, y)_{w_i}/\sum_{j \neq i} f(y_j, y)_{w_i}$. Our adaptive weights vary with the inner variation $\{(y_i - y)^2 | t_i \neq t\}$ (decreased or increased), which adjusts (shrinking or magnifying) the distance of points across each time step and achieves an adaptive feature-scale transformation. Specifically, the minor variation gets further shrunk at a large feature scale, magnified at a small feature scale, and vice versa. Concerning random components, global attention can serve as an average smoothing method to help filter small perturbations. As for outliers, their large margin against regular items leads to minor weights, which eliminates the interference of outliers. Especially when the sample $(t_i, y_i)$ comes to be an outlier, this structure brushes itself aside. Thus, the smoothing filter attention mechanism filters out random components and dynamically adjusts feature scales. This way, we can dynamically transform non-stationary time series according to different scales, which accesses time series under comprehensive sights.

*Difference attention module to discover stable regularities.* The difference attention module calculates the internal connections among stable shifted features to discover stable regularities within the non-stationary time series and thereby overcomes the interference of uneven distributions. Concretely, as shown in Fig. 2b, this module includes the difference and CumSum operations at both ends of the self-attention mechanism[35], which interconnects the shift across each time step to capture internal connections within non-stationary time series. The difference operation separates the shifts from the long-term trends, where the shift refers to the minor difference in the trends between adjacent time steps. Considering trends lead the data distribution to change over time, the difference operation makes the time series stable

and varies around a fixed mean level with minor distribution shifts. Subsequently, we use a self-attention mechanism to interconnect shifts, which captures the temporal dependencies within the time series variation. Last, we employ a CumSum operation to accumulate shifted features and generate a non-stationary time series conforming to the discovered regularities.

*Modeling and generating non-stationary time series in Diviner framework.* The smoothing filter attention mechanism filters out random components and dynamically adjusts the feature scale. Subsequently, the difference attention module calculates internal connections and captures the stable regularity within the time series at the corresponding scale. Cascading these two modules, one Diviner block can discover stable regularities within non-stationary time series at one scale. Then, we stack Diviner blocks in a multilayer structure to achieve multi-scale transformation layers and capture multi-scale stable features from non-stationary time series. Such a multilayer structure is organized in an encoder-decoder architecture with asymmetric input lengths for efficient data utilization. The encoder takes a long historical series to embed trends, and the decoder receives a relatively short time series. With the cross-attention between the encoder and decoder, we can pair the latest time series with pertinent variation patterns from long historical series and make inferences about future trends, improving calculation efficiency and reducing redundant historical information. The point is that the latest time series is more conducive to anticipating the immediate future than the remote-past time series, where the correlation across time steps generally degrades with the length of the interval[53–57]. Additionally, we design a generator to obtain prediction results in one step to avoid dynamic cumulative error problems[39]. The generator is built with CovNet sharing parameters throughout each time step based on the linear projection generator[39,58,59], which saves hardware resources. These techniques enable deep learning methods to model non-stationary time series with multi-scale stable features and produce forecasting results in a generative

**Table 1 Time-series forecasting results on the 5G traffic network dataset.**

| Models | | Diviner | | | Autoformer | | | Informer | | | Transformer | | | NBeats | | |
|---|---|---|---|---|---|---|---|---|---|---|---|---|---|---|---|---|
| Metric | | MSE | MAE | MASE | MSE | MAE | MASE | MSE | MAE | MASE | MSE | MAE | MASE | MSE | MAE | MASE |
| NPT-1 | 96 | **0.256** | **0.340** | **1.391** | 0.456 | 0.511 | 2.090 | 0.264 | 0.349 | 1.427 | 0.259 | 0.333 | 1.362 | 0.491 | 0.509 | 2.082 |
| | 288 | **0.277** | **0.379** | **1.598** | 0.431 | 0.499 | 2.104 | 0.611 | 0.590 | 2.488 | 0.376 | 0.445 | 1.876 | 0.624 | 0.694 | 2.927 |
| | 672 | **0.263** | **0.367** | **1.601** | 0.446 | 0.522 | 2.278 | 1.680 | 0.885 | 3.862 | 0.365 | 0.437 | 1.907 | 0.680 | 0.615 | 2.684 |
| | 1344 | **0.275** | **0.367** | **1.585** | 0.400 | 0.467 | 2.017 | 1.307 | 0.923 | 3.987 | 0.448 | 0.462 | 1.996 | 0.883 | 0.692 | 2.989 |
| | 2880 | **0.318** | **0.390** | **1.613** | 0.674 | 0.629 | 2.601 | 1.590 | 1.050 | 4.343 | 0.811 | 0.652 | 2.697 | 1.257 | 0.844 | 3.491 |
| NPT-2 | 96 | **0.370** | **0.405** | **1.800** | 0.605 | 0.603 | 2.681 | 0.760 | 0.646 | 2.870 | 0.458 | 0.470 | 2.088 | 0.539 | 0.476 | 2.116 |
| | 288 | **0.394** | **0.431** | **1.977** | 0.579 | 0.607 | 2.786 | 1.131 | 0.826 | 3.788 | 0.415 | 0.454 | 2.082 | 0.589 | 0.541 | 2.481 |
| | 672 | **0.484** | **0.462** | **2.074** | 0.541 | 0.525 | 2.357 | 1.149 | 0.861 | 3.864 | 0.548 | 0.546 | 2.453 | 0.734 | 0.598 | 2.685 |
| | 1344 | **0.314** | **0.372** | **1.814** | 0.437 | 0.472 | 2.301 | 1.129 | 0.858 | 4.181 | 0.705 | 0.593 | 2.889 | 0.583 | 0.532 | 2.593 |
| | 2880 | **0.378** | **0.390** | **1.861** | 0.750 | 0.644 | 3.072 | 1.342 | 0.935 | 4.457 | 0.458 | 0.470 | 2.240 | 0.934 | 0.725 | 3.459 |
| NPT-3 | 96 | **0.177** | **0.323** | **1.672** | 0.272 | 0.401 | 2.076 | 0.664 | 0.656 | 3.397 | 0.300 | 0.415 | 2.150 | 0.227 | 0.347 | 1.797 |
| | 288 | **0.193** | **0.301** | **1.558** | 0.579 | 0.607 | 3.144 | 0.880 | 0.721 | 3.736 | 0.458 | 0.478 | 2.478 | 0.486 | 0.498 | 2.579 |
| | 672 | **0.187** | **0.305** | **1.599** | 0.541 | 0.525 | 2.753 | 0.931 | 0.771 | 4.044 | 0.327 | 0.409 | 2.147 | 0.455 | 0.488 | 2.558 |
| | 1344 | **0.204** | **0.335** | **1.822** | 0.437 | 0.472 | 2.569 | 1.023 | 0.831 | 4.520 | 0.362 | 0.434 | 2.363 | 0.622 | 0.575 | 3.128 |
| | 2880 | **0.240** | **0.350** | **1.756** | 0.750 | 0.644 | 3.228 | 1.196 | 0.922 | 4.622 | 0.362 | 0.434 | 2.177 | 0.816 | 0.673 | 3.374 |

The traffic forecast accuracy is assessed by MSE, MAE, and MASE. $\text{MSE} = \frac{1}{n}\sum_{i=1}^{n}(\mathbf{y}_i - \hat{\mathbf{y}}_i)^2$, $\text{MAE} = \frac{1}{n}\sum_{i=1}^{n}|\mathbf{y}_i - \hat{\mathbf{y}}_i|$, $\text{MASE} = \frac{\frac{1}{n}\sum_{i=1}^{n}|\mathbf{y}_i - \hat{\mathbf{y}}_i|}{\frac{1}{n-1}\sum_{i=2}^{n}|\mathbf{y}_i - \mathbf{y}_{i-1}|}$, where $\hat{\mathbf{y}} \in \mathbb{R}^n$ denotes the forecast and $\mathbf{y} \in \mathbb{R}^n$ denotes the ground truth. All datasets were standardized using the mean and standard deviation values of the training set. The best predictive performance over the comparison is shown in bold.

paradigm, which is an attempt to tackle long-term time series prediction problems.

**Performance of the 5G network traffic forecasting.** To validate the effectiveness of the proposed techniques, we collect extensive NPTs from China Unicom. The NPT datasets include data recorded every 15 minutes for the whole 2021 year from three groups of real-world metropolitan network traffic ports {NPT-1, NPT-2, NPT-3}, where each sub-dataset contains {18, 5, 5} ports, respectively. We split them chronologically with a 9:1 proportion for training and testing. In addition, we prepare 16 network ports for parameter-searching. The main difficulties lie in the explicit shift of the distribution and numerous outliers. And this Section elaborates on the comprehensive comparison of our model with prediction-based and growth-rate-based models in applying 5G network traffic forecasting.

Experiment 1: We first compare Diviner to other time series prediction-based methods, we note these baseline models as Baselines-T for clarity. Baselines-T include traditional models ARIMA[19,20] and Prophet[26]; classic machine learning model LSTMa[60]; deep learning-based models Transformer[35], Informer[39], Autoformer[42], and NBeats[61]. These models are required to predict the entire network traffic series {1, 3, 7, 14, 30} days, aligned with {96, 288, 672, 1344, 2880} prediction spans ahead in Table 1, and inbits is the target feature. In terms of the evaluation, although the MAE, MSE, and MASE predictive accuracy generally decrease with prediction intervals, the degradation rate varies between models. Therefore, we introduce an exponential velocity indicator to measure the rate of accuracy degradation. Specifically, given time spans $[t_1, t_2]$ and the corresponding MSE, MAE, and MASE errors, we have the following:

$$\text{dMSE}_{t_1}^{t_2} = \left( \sqrt[t_2 - t_1]{\text{MSE}_{t_2}/\text{MSE}_{t_1}} - 1 \right) \times 100\%, \qquad (2)$$

$$\text{dMAE}_{t_1}^{t_2} = \left( \sqrt[t_2 - t_1]{\text{MAE}_{t_2}/\text{MAE}_{t_1}} - 1 \right) \times 100\%, \qquad (3)$$

$$\text{dMASE}_{t_1}^{t_2} = \left( \sqrt[t_2 - t_1]{\text{MASE}_{t_2}/\text{MASE}_{t_1}} - 1 \right) \times 100\%, \qquad (4)$$

where $\text{dMSE}_{t_1}^{t_2}, \text{dMAE}_{t_1}^{t_2}, \text{dMASE}_{t_1}^{t_2} \in \mathbb{R}$. Concerning the close experimental results between {NPT-1, NPT-2, and NPT-3}, we focus mainly on the result of the NPT-1 dataset, and the

experimental results are summarized in Table 1. Although there exist quantities of outliers and frequent oscillations in the NPT dataset, Diviner achieves a 38.58% average MSE reduction ($0.451 \rightarrow 0.277$) and a 20.86% average MAE reduction ($0.465 \rightarrow 0.368$) based on the prior art. In terms of the scalability to different prediction spans, Diviner has a much lower $\text{dMSE}_1^{30}$ ($4.014\% \rightarrow 0.750\%$) and $\text{dMAE}_1^{30}$ ($2.343\% \rightarrow 0.474\%$) than the prior art, which exhibits a slight performance degradation with a substantial improvement in predictive robustness when the prediction horizon becomes longer. The degradation rates and predictive performance of all baseline approaches have been provided in Supplementary Table S1 regarding to the space limitation.

The experiments on NPT-2 and NPT-3 shown in Supplementary Data 1 reproduce the above results, where Diviner can support accurate long-term network traffic prediction and exceed current art involving accuracy and robustness by a large margin. In addition, we have the following results by sorting the comprehensive performances (obtained by the average MASE errors) of the baselines established with the Transformer framework: Diviner > Autoformer > Transformer > Informer. This order aligns with the non-stationary factors considered in these models and verifies our proposal that incorporating non-stationarity promotes neural networks' adaptive abilities to model time series, and the modeling multi-scale non-stationarity other breaks through the ceiling of prediction abilities for deep learning models.

Experiment 2: The second experiment compares Diviner with two other industrial methods, which aim to predict the capacity utilization of inbits and outbits with historical growth rates. The experiment shares the same network port traffic data as in Experiment 1, while the split ratio is changed to 3:1 chronologically for a longer prediction horizon. Furthermore, we use a long construction cycle of {30, 60, 90} days (aligned with {2880, 5760, 8640} time steps) to ensure the validity of such growth-rate-based methods for the law of large numbers. Here we first define capacity utilization mathematically:

Given a fixed bandwidth $B \in \mathbb{R}$ and the traffic flow of the $k$th construction cycles $\widetilde{\mathbf{X}}(k) = \begin{bmatrix} \tilde{\mathbf{x}}_{kC+1} & \tilde{\mathbf{x}}_{kC+2} & \cdots & \tilde{\mathbf{x}}_{(k+1)C} \end{bmatrix}$, $\widetilde{\mathbf{X}}(k) \in \mathbb{R}^{T \times C}$, where $\tilde{\mathbf{x}}_i \in \mathbb{R}^T$ is a column vector of length $T$ representing the time series per day and $C$ denotes the number of days in one construction cycle. Then the capacity utilization (CU)

**Table 2 Long-term (1–3 months) capacity utilization forecasting results on the NPT dataset.**

| Models | | Diviner | | Baseline-A | | Baseline-M | |
|---|---|---|---|---|---|---|---|
| MAE, MASE | | Inbits | Outbits | Inbits | Outbits | Inbits | Outbits |
| 7D2NPT-1 | 2880 | **0.390, 2.388** | **0.552, 3.420** | 0.792, 4.851 | 0.874, 5.415 | 2.888, 17.690 | 5.693, 35.274 |
| | 5760 | **0.705, 4.351** | **0.899, 6.494** | 0.870, 5.371 | 0.948, 6.848 | 31.490, 194.431 | 17.602, 127.16 |
| | 8640 | **0.640, 3.425** | **0.694, 4.172** | 0.877, 4.693 | 1.012, 6.084 | 21.220, 113.567 | 14.487, 87.096 |
| 7D2NPT-2 | 2880 | **0.352, 1.992** | **0.525, 2.574** | 0.911, 5.159 | 0.907, 4.442 | 2.644, 14.959 | 2.112, 10.341 |
| | 5760 | **0.508, 2.817** | **0.605, 2.905** | 1.037, 5.751 | 1.037, 4.980 | 8.464, 46.936 | 5.515, 26.470 |
| | 8640 | **0.814, 4.363** | **0.642, 3.113** | 1.169, 6.265 | 1.189, 5.766 | 8.323, 44.602 | 7.048, 34.165 |
| 7D2NPT-3 | 2880 | **0.398, 2.232** | **0.413, 2.987** | 0.945, 5.291 | 0.719, 5.192 | 4.993, 27.947 | 5.222, 37.698 |
| | 5760 | **0.769, 4.713** | **0.719, 4.911** | 1.048, 6.425 | 0.817, 5.580 | 6.895, 42.245 | 15.078, 102.973 |
| | 8640 | **0.621, 3.611** | **0.604, 4.062** | 0.968, 5.622 | 0.829, 5.567 | 5.169, 30.023 | 29.547, 198.393 |

The long-term capacity utilization forecasting results on the NPT dataset were evaluated using MAE and MASE error metrics. All datasets were standardized using the mean and standard deviation values of the training set. The best predictive performance over the comparison is shown in bold.

of the $k$th construction cycle is defined as follows:

$$CU(k) = \frac{\| \widetilde{\mathbf{X}}(k)\|_{m1}}{BCT}, \quad (5)$$

where $CU(k) \in \mathbb{R}$. As shown in the definition, capacity utilization is directly related to network traffic, so a precise network traffic prediction leads to a quality prediction of capacity utilization. We compare the proposed predictive method with two commonly used moving average growth rate predictive methods in the industry, the additive and multiplicative moving average growth rate predictive methods. For clarity, we note the additive method as Baseline-A and the multiplicative method as Baseline-M. Baseline-A calculates an additive growth rate with the difference of adjacent construction cycles. Given the capacity utilization of the last two construction cycles $CU(k−1)$, $CU(k−2)$, we have the following:

$$\widehat{CU}_A(k) = 2\,CU(k−1) − CU(k−2). \quad (6)$$

Baseline-M calculates a multiplicative growth rate with the quotient of adjacent construction cycles. Given the capacity utilization of the last two construction cycles $CU(k−1)$, $CU(k−2)$, we have the following:

$$\widehat{CU}_M(k) = \frac{CU(k−1)}{CU(k−2)}\,CU(k−1). \quad (7)$$

Different from the above two baselines, we calculate the capacity utilization of the network with the network traffic forecast. Given the network traffic of the last $K$ construction cycles $\widetilde{\mathbf{X}} = \begin{bmatrix} \tilde{\mathbf{x}}_{(k−K)C+1} & \cdots & \tilde{\mathbf{x}}_{(k−K+1)C} & \cdots & \tilde{\mathbf{x}}_{(k−1)C} & \cdots & \tilde{\mathbf{x}}_{kC} \end{bmatrix}$, we have the following:

$$\widetilde{\mathcal{X}}(k) = \text{Diviner}(\widetilde{\mathcal{X}}), \quad (8)$$

$$\widehat{CU}_D(k) = \frac{\| \widetilde{\mathcal{X}}(k)\|_{m1}}{BCT}. \quad (9)$$

We summarize the experimental results in Table 2. Concerning the close experimental results between {NPT-1, NPT-2, and NPT-3} shown in, we focus mainly on the result of the NPT-1 dataset, which has the most network traffic ports. Diviner achieves a substantial reduction of 31.67% MAE ($0.846 \rightarrow 0.578$) on inbits and a reduction of 24.25% MAE ($0.944 \rightarrow 0.715$) on outbits over Baseline-A. An intuitive explanation is that the growth-rate-based methods extract particular historical features but lack adaptability. We notice that Baseline-A has a much better performance of $0.045\times$ average inbits-MAE and $0.074\times$ average outbits-MAE over Baseline-M. This result suggests that network traffic tends to increase linearly rather than exponentially. Nevertheless, there

remain inherent multi-scale variations of network traffic series, so Diviner still exceeds the Baseline-A, suggesting the necessity of applying deep learning models such as Diviner to discover nonlinear latent regularities within network traffic.

When analyzing the results of these two experiments jointly, we present that Diviner possesses a relatively low degradation rate for a prediction of 90 days, $\text{dMASE}_1^{90} = 1.034\%$. In contrast, the degradation rate of the prior art comes to $\text{dMASE}_1^{30} = 2.343\%$ for a three-times shorter prediction horizon of 30 days. Furthermore, considering diverse network traffic patterns in the provided datasets (about 50 ports), the proposed method can deal with a wide range of non-stationary time series, validating its applicability without modification. These experiments witness Diviner's success in providing quality long-term network traffic forecasting and extending the effective prediction spans of deep learning models for up to three months.

**Application on other real-world datasets**. We validate our method on benchmark datasets for the weather (WTH), electricity transformer temperature (ETT), electricity (ECL), and exchange (Exchange). We summarize the experimental results in Table 3. We follow the standard protocol and divide them into training, validation, and test sets in chronological order with a proportion of 7:1:2 unless otherwise specified. Due to the space limitation, the complete experimental results are shown in Supplementary Data 2.

*Weather temperature prediction.* The WTH dataset[42] records 21 meteorological indicators for Jena 2020, including air temperature and humidity, and WetBulbFarenheit is the target. This dataset is finely quantified to the 10-min level, which means that there are 144 steps for one day and 4320 steps for one month, thereby challenging the capacity of models to process long sequences. Among all baselines, NBeats and Informer have the lowest error in terms of MSE and MAE metrics, respectively. However, we notice a contrast between these two models when extending prediction spans. Informer degrades precipitously when the prediction spans increase from 2016 to 4032 (MAE:$0.417 \rightarrow 0.853$), but on the contrary, NBeats gains a performance improvement (MAE:$0.635 \rightarrow 0.434$). We attribute this to a trade-off of pursuing context and texture. Informer has an advantage over the texture in the short-term case. Still, it needs to capture the context dependency of the series considering the length of input history series should extend in pace with prediction spans and vice versa. As for Diviner, it achieves a remarkable 29.30% average MAE reduction ($0.488 \rightarrow 0.345$) and 41.54% average MSE reduction ($0.491 \rightarrow 0.287$) over both Informer and NBeats. Additionally, Diviner gains a low degradation rate of $\text{dMSE}_1^{30} = 0.439\%$,

**Table 3 Time-series forecasting results on other real-world datasets.**

| Models | | Diviner | | | Autoformer | | | Informer | | | Transformer | | | NBeats | | |
|---|---|---|---|---|---|---|---|---|---|---|---|---|---|---|---|---|
| Metric | | MSE | MAE | MASE | MSE | MAE | MASE | MSE | MAE | MASE | MSE | MAE | MASE | MSE | MAE | MASE |
| WTH | 144 | **0.280** | **0.341** | **7.072** | 0.373 | 0.440 | 9.125 | 0.359 | 0.401 | 8.316 | 0.448 | 0.484 | 10.038 | 0.508 | 0.590 | 12.236 |
| | 432 | **0.333** | **0.392** | **8.135** | 0.402 | 0.445 | 9.235 | 0.374 | 0.431 | 8.944 | 0.407 | 0.470 | 9.754 | 0.427 | 0.501 | 10.397 |
| | 1008 | **0.273** | **0.328** | **6.806** | 0.663 | 0.613 | 12.720 | 0.344 | 0.387 | 8.030 | 0.535 | 0.514 | 10.666 | 0.406 | 0.490 | 10.167 |
| | 2016 | **0.233** | **0.306** | **6.348** | 1.857 | 1.019 | 21.140 | 0.367 | 0.417 | 8.651 | 0.367 | 0.417 | 8.651 | 0.757 | 0.635 | 13.173 |
| | 4032 | **0.318** | **0.358** | **5.832** | 1.016 | 0.853 | 13.897 | 1.251 | 0.806 | 13.131 | 0.876 | 0.616 | 10.035 | 0.361 | 0.434 | 7.070 |
| ETTh1 | 24 | **0.058** | **0.183** | **4.174** | 0.093 | 0.234 | 5.338 | 0.098 | 0.247 | 5.634 | 0.468 | 0.599 | 13.664 | 0.157 | 0.269 | 6.136 |
| | 48 | **0.071** | **0.203** | **4.629** | 0.089 | 0.229 | 5.222 | 0.158 | 0.319 | 7.274 | 0.369 | 0.524 | 11.950 | 0.146 | 0.292 | 6.659 |
| | 168 | **0.119** | **0.262** | **5.977** | 0.148 | 0.280 | 6.387 | 0.183 | 0.346 | 7.893 | 0.478 | 0.618 | 14.099 | 0.494 | 0.536 | 12.228 |
| | 336 | **0.114** | **0.268** | **6.031** | 0.183 | 0.344 | 7.742 | 0.222 | 0.387 | 8.710 | 0.235 | 0.417 | 9.385 | 0.411 | 0.494 | 11.118 |
| | 720 | **0.157** | **0.322** | **7.195** | 0.201 | 0.364 | 8.133 | 0.269 | 0.435 | 9.720 | 0.261 | 0.445 | 9.943 | 1.257 | 0.844 | 18.859 |
| ETTm1 | 24 | **0.157** | **0.322** | **7.195** | 0.201 | 0.364 | 8.133 | 0.269 | 0.435 | 9.720 | 0.261 | 0.445 | 9.943 | 1.257 | 0.844 | 18.859 |
| | 48 | **0.023** | **0.119** | **5.159** | 0.092 | 0.250 | 10.838 | 0.069 | 0.230 | 9.971 | 0.288 | 0.481 | 20.853 | 1.584 | 1.220 | 52.892 |
| | 96 | **0.044** | **0.162** | **6.957** | 0.063 | 0.198 | 8.503 | 0.194 | 0.372 | 15.975 | 0.264 | 0.450 | 19.325 | 1.352 | 1.106 | 47.497 |
| | 288 | **0.078** | **0.209** | **9.375** | 0.096 | 0.245 | 10.990 | 0.401 | 0.554 | 24.850 | 0.230 | 0.410 | 18.391 | 0.628 | 0.621 | 27.856 |
| | 672 | **0.071** | **0.211** | **9.323** | 0.117 | 0.276 | 12.195 | 0.512 | 0.644 | 28.456 | 0.379 | 0.540 | 23.861 | 0.361 | 0.480 | 21.210 |
| ETTh2 | 24 | **0.072** | **0.203** | **4.281** | 0.131 | 0.281 | 5.927 | 0.093 | 0.240 | 5.062 | 0.608 | 0.653 | 13.773 | 0.167 | 0.318 | 6.707 |
| | 48 | **0.109** | **0.252** | **5.209** | 0.143 | 0.284 | 5.871 | 0.155 | 0.314 | 6.491 | 0.758 | 0.740 | 15.298 | 0.264 | 0.392 | 8.104 |
| | 168 | **0.206** | **0.352** | **6.070** | 0.254 | 0.399 | 6.881 | 0.232 | 0.389 | 6.708 | 0.425 | 0.528 | 9.105 | 0.525 | 0.548 | 9.450 |
| | 336 | **0.220** | **0.373** | **5.792** | 0.262 | 0.403 | 6.258 | 0.263 | 0.417 | 6.475 | 0.324 | 0.461 | 7.158 | 0.750 | 0.655 | 10.171 |
| | 720 | **0.202** | **0.368** | **5.435** | 0.579 | 0.621 | 9.172 | 0.277 | 0.431 | 6.365 | 0.270 | 0.423 | 6.247 | 0.816 | 0.682 | 10.073 |
| ECL | 168 | 0.265 | **0.361** | **2.315** | 0.385 | 0.458 | 2.937 | 0.447 | 0.503 | 3.225 | 0.587 | 0.561 | 3.597 | **0.225** | 0.363 | 2.327 |
| | 336 | 0.295 | 0.395 | 2.602 | 0.462 | 0.496 | 3.267 | 0.489 | 0.528 | 3.478 | 0.683 | 0.64 | 4.215 | **0.237** | **0.359** | **2.364** |
| | 720 | **0.303** | **0.409** | **2.544** | 1.349 | 0.907 | 5.643 | 0.54 | 0.571 | 3.552 | 0.482 | 0.527 | 3.278 | 0.367 | 0.482 | 2.998 |
| | 960 | **0.427** | **0.489** | **3.849** | 1.263 | 0.920 | 7.242 | 0.582 | 0.608 | 4.786 | 0.644 | 0.597 | 4.699 | 0.457 | 0.540 | 4.250 |
| Exchange | 10 | **0.147** | **0.282** | **2.867** | 0.163 | 0.315 | 3.203 | 4.896 | 2.124 | 21.601 | 6.926 | 2.553 | 25.964 | 0.804 | 0.701 | 7.129 |
| | 20 | **0.273** | **0.421** | **3.160** | 0.423 | 0.540 | 4.054 | 6.318 | 2.443 | 18.341 | 6.759 | 2.524 | 18.949 | 1.166 | 0.939 | 7.049 |
| | 30 | **0.399** | **0.506** | **3.132** | 0.857 | 0.799 | 4.945 | 5.388 | 2.253 | 13.945 | 7.307 | 2.635 | 16.31 | 1.521 | 1.105 | 6.839 |
| | 60 | **0.619** | **0.669** | **4.265** | 0.911 | 0.776 | 4.948 | 9.886 | 3.067 | 19.557 | 8.455 | 2.840 | 18.109 | 3.299 | 1.670 | 10.648 |
| Solar | 144 | **0.348** | **0.326** | **7.461** | 0.431 | 0.485 | 11.091 | 0.365 | 0.362 | 8.290 | 0.546 | 0.513 | 11.742 | 0.351 | 0.371 | 8.487 |
| | 288 | **0.312** | **0.331** | **8.355** | 0.437 | 0.477 | 12.035 | 0.405 | 0.397 | 10.007 | 0.368 | 0.368 | 9.289 | 0.345 | 0.356 | 8.988 |
| | 720 | **0.315** | **0.342** | **8.793** | 0.400 | 0.525 | 13.497 | 0.577 | 0.537 | 13.803 | 0.339 | 0.441 | 11.352 | 0.350 | 0.357 | 9.176 |
| | 864 | **0.310** | **0.297** | **7.053** | 0.546 | 0.607 | 14.423 | 0.994 | 0.897 | 21.299 | 0.813 | 0.478 | 11.367 | 0.349 | 0.357 | 8.488 |
| Traffic | 168 | **0.156** | **0.259** | **0.835** | 0.431 | 0.485 | 1.561 | 1.814 | 1.159 | 3.729 | 0.750 | 0.644 | 2.071 | 0.509 | 0.528 | 1.700 |
| | 336 | **0.158** | **0.261** | **0.847** | 0.437 | 0.477 | 1.548 | 1.799 | 1.153 | 3.738 | 0.629 | 0.573 | 1.857 | 0.517 | 0.529 | 1.714 |
| | 720 | **0.318** | **0.437** | **1.457** | 0.400 | 0.525 | 1.751 | 1.817 | 1.150 | 3.836 | 0.671 | 0.604 | 2.014 | 0.526 | 0.533 | 1.779 |
| | 960 | **0.277** | **0.397** | **1.299** | 0.546 | 0.607 | 1.986 | 1.821 | 1.165 | 3.809 | 1.950 | 1.116 | 3.649 | 0.523 | 0.532 | 1.740 |

The model's predictive accuracy is assessed by MSE, MAE, and MASE. All datasets were standardized using the mean and standard deviation values of the training set. The best and suboptimal predictive performance over the comparison is shown in bold and italics, respectively.

$dMAE_1^{30} = 0.167\%$ showing its ability to harness historical information within time series. The predictive performances and degradation rates of all baseline approaches have been provided in Supplementary Table S2. Our model can synthesize context and texture to balance both short-term and long-term cases, ensuring its accurate and robust long-term prediction.

*Electricity transformer temperature prediction.* The ETT dataset contains two-year data with six power load features from two counties in China, and oil temperature is our target. Its split ratio of training/validation/test set is 12/4/4 months[39]. The ETT data set is divided into two separate datasets at the 1-h {ETT$h_1$, ETT$h_2$} and 15-minute levels ETT$m_1$. Therefore, we can study the performance of the models under different granularities, where the prediction steps {96, 288, 672} of ETT$m_1$ align with the prediction steps {24, 48, 168} of ETT$h_1$. Our experiments show that Diviner achieves the best performance in both cases. Although in the hour-level case, Diviner outperforms the baselines with the closest MSE and MAE to Autoformer (MSE: 0.110 → 0.082, MAE: 0.247 → 0.216). When the hour-level granularity turns to a minute-level case, Diviner outperforms Autoformer by a large margin (MSE:0.092 → 0.064, MAE:0.239 → 0.194). The predictive performances and the granularity change when the hour-level granularity turns into the minute-level granularity of all baseline approaches have been provided in Supplementary Table S3. These demonstrate the capacity of the Diviner in processing time series of different granularity. Furthermore, the granularity is also a manifestation of scale. These results demonstrate that modeling multi-scale features is conducive to dealing with time series of different granularity.

*Consumer electricity consumption prediction.* The ECL dataset records the two-year electricity consumption of 321 clients, which is converted into hour-level consumption owing to the missing data, and MT-320 is the target feature[62]. We predict different time horizons of {7, 14, 30, 40} days, aligned with {168, 336, 720, 960} prediction steps ahead. Next, we analyze the experimental results according to the prediction spans (≤360 as short-term prediction, ≥360 as long-term prediction). NBeats achieves the best forecasting performance for short-term electricity consumption prediction, while Diviner surpasses it in the long-term prediction case. The short-term and long-term performance of all approaches has been provided in Supplementary Table S4. Statistically, the proposed method outperforms the best baseline (NBeats) by decreasing 17.43% MSE (0.367 → 0.303), 15.14% MAE (0.482 → 0.409) at 720 steps ahead, and 6.56% MSE (0.457 → 0.427) at 9.44% MAE (0.540 → 0.489) at 960 steps ahead. We attribute this to scalability, where different models converge to perform similarly in the short-term case, but their differences emerge when the prediction span becomes longer.

*Gold price prediction.* The Exchange dataset contains 5-year closing prices of a troy ounce of gold in the US recorded daily from 2016 to 2021. Due to the high-frequency fluctuation of the market price, the predictive goal is to predict its general trend reasonably (https://www.lbma.org.uk). To this end, we perform a long-term prediction of {10, 20, 30, 60} days. The experimental results clearly show apparent performance degrades for most baseline models. Given a history of 90 days, only Autoformer and Diviner can predict with MAE and MSE errors lower than 1 when the prediction span is 60 days. However, Diviner still outperforms other methods with a 38.94% average MSE reduction (0.588 → 0.359) and a 22.73% average MSE reduction (0.607 → 0.469) and achieves the best forecast performance. The predictive performance of all baseline approaches has been provided in Supplementary Table S5. These results indicate the adaptability of Diviner to the rapid evolution of financial markets and its reasonable extrapolation, considering that it is generally difficult to predict the financial system.

*Solar energy production prediction.* The solar dataset contains the 10-minute level 1 year (2006) solar power production data of 137 PV plants in Alabama State, and PV-136 is the target feature (http://www.nrel.gov). Given that the amount of solar energy produced daily is generally stable, conducting a super long-term prediction is unnecessary. Therefore, we set the prediction horizon to {1, 2, 5, 6} days, aligned with {144, 288, 720, 864} prediction steps ahead. Furthermore, this characteristic of solar energy means that its production series tend to be stationary, and thereby the comparison of the predictive performances between different models on this dataset presents their basic series modeling abilities. Concretely, considering the MASE error can be used to assess the model's performance on different series, we calculate and sort each model's average MASE error under different prediction horizon settings to measure the time series modeling ability (provided in Supplementary Table S6). The results are as follows: Diviner > NBeats > Transformer > Autoformer > Informer > LSTM, where Diviner surpasses all Transformer-based models in the selected baselines. Provided that the series data is not that non-stationary, the advantages of Autoformer's modeling time series non-stationarity are not apparent. At the same time, capturing stable long- and short-term dependencies is still effective.

*Road occupancy rate prediction.* The Traffic dataset contains hourly 2-year (2015–2016) road occupancy rate collected from 862 sensors on San Francisco Bay area freeways by the California Department of Transportation, where sensor-861 is the target feature (http://pems.dot.ca.gov). The prediction horizon is set to {7, 14, 30, 40} days, aligned with {168, 336, 720, 960} prediction steps ahead. Considering the road occupancy rate tends to have a weekly cycle, we use this dataset to compare different networks' ability to model the temporal cycle. During the comparison, we mainly focus on the following two groups of deep learning models: group-1 takes the non-stationary specialization of time series into account (Diviner, Autoformer), and group-2 does not employ any time-series-specific components (Transformer, Informer, LSTMa). We find that group-1 gains a significant performance improvement over group-2, which suggests the necessity of modeling non-stationarity. As for the proposed Diviner model, it achieves a 27.64% MAE reduction (0.604 → 0.437) to the Transformer model when forecasting 30-day road occupancy rates. Subsequently, we conduct an intra-group comparison for group-1, where Diviner still gains an average 35.37% MAE reduction (0.523 → 0.338) to Autoformer. The predictive performance of all approaches has been provided in Supplementary Table S7. We attribute this to Diviner's multiple-scale modeling of non-stationarity, while the trend-seasonal decomposition of Autoformer merely reflects time series variation at particular scales. These experimental results demonstrate that Diviner is competent in predicting time series data with cycles.

## Discussion

We study the long-term 5G network traffic prediction problem by modeling non-stationarity with deep learning techniques. Although some literature[63–65] in the early stage argues that the probabilistic traffic forecast under uncertainty is more suitable for the varying network traffic than a concrete forecast produced by time series models, the probabilistic traffic forecast and the concrete traffic forecast share the same historical information in essence. Moreover, the development of time series forecasting techniques these years has witnessed a series of works employing time series forecasting

techniques for practical applications such as bandwidth management[14,15], resource allocation[16], and resource provisioning[17], where the time series prediction-based methods can provide detailed network traffic forecast. However, existing time series forecasting methods suffer a severe performance degeneration since the long-term prediction horizon exposes the non-stationarity of time series, which raises several challenges: (a) Multi-scale temporal variations. (b) Random factors. (c) Data Distribution Shift.

Therefore, this paper attempts to challenge the problem of achieving a precise long-term prediction for non-stationary time series. We start from the fundamental property of time series non-stationarity and introduce deep stationary processes into a neural network, which models multi-scale stable regularities within non-stationary time series. We argue that capturing the stable features is a recipe for generating non-stationary forecasts conforming to historical regularities. The stable features enable networks to restrict the latent space of time series, which deals with varying distribution problems. Extensive experiments on network traffic prediction and other real-world scenarios demonstrate its advances over existing prediction-based models. Its advantages are summarized as follows. (a) Diviner brings a salient improvement on both long- and short-term prediction and achieves state-of-the-art performance. (b) Diviner can perform robustly regardless of the selection of prediction span and granularity, showing great potential for long-term forecasting. (c) Diviner maintains a strong generalization in various fields. The performance of most baselines might degrade precipitously in some or other areas. In contrast, our model distinguishes itself for consistent performance on each benchmark.

This work explores an avenue to obtain detailed and precise long-term 5G network traffic forecasts, which can be used to calculate the time network traffic might overflow the capacity and helps operators formulate network construction schemes months in advance. Furthermore, Diviner generates long-term network traffic forecasts at the minute level, facilitating its broader applications for resource provisioning, allocating, and monitoring. Decision-makers can harness long-term predictions to allocate and optimize network resources. Another practical application is to achieve an automatic network status monitoring system, which automatically alarms when real network traffic exceeds a permitted range around predictions. This system supports targeted port-level early warning and assists workers in troubleshooting in time, which can bring substantial efficiency improvement considering the tens of millions of network ports running online. In addition to 5G networks, we have expanded our solution to broader engineering fields such as electricity, climate, control, economics, energy, and transportation. Predicting oil temperature can help prevent the transformer from overheating, which affects the insulation life of the transformer and ensures proper operation[66,67]. In addition, long-term meteorological prediction helps to select and seed crops in agriculture. As such, we can discover unnoticed regularities within historical series data, which might bring opportunities to traditional industries.

One limitation of our proposed model is that it suffers from critical transitions of data patterns. We attribute this to external factors, whose information is generally not included in the measured data[53,55,68]. Our method is helpful in the intrinsic regularity discovery within the time series but cannot predict patterns not previously recorded in the real world. Alternatively, we can use dynamic network methods[69–71] to detect such critical transitions in the time series[53]. Furthermore, the performance of Diviner might be similar to other deep learning models if given a few history series or in the short-term prediction case. The former contains insufficient information to be exploited, and the short-term prediction needs more problem scalability, whereas the

advantages of our model become apparent in long-term forecasting scenarios.

## Methods

**Preliminaries**. We denote the original form of the time-series data as $\mathbf{X} = \begin{bmatrix} x_1 & x_2 & \dots & x_n \end{bmatrix}, x_i \in \mathbb{R}$. The original time series data $\mathbf{X}$ is reshaped to a matrix form as $\widetilde{\mathbf{X}} = \begin{bmatrix} \tilde{\mathbf{x}}_1 & \tilde{\mathbf{x}}_2 & \dots & \tilde{\mathbf{x}}_K \end{bmatrix}$, where $\tilde{\mathbf{x}}_i$ is a vector of length $T$ with the time series data per day/week/month/year, $K$ denotes the number of days/weeks/months/years, $\tilde{\mathbf{x}}_i \in \mathbb{R}^T$. After that, we can represent *the seasonal pattern* as $\tilde{\mathbf{x}}_i$ and use its variation between adjacent time steps to model *trends*, shown as the following:

$$
\begin{aligned}
\tilde{\mathbf{x}}_{t_2} &= \tilde{\mathbf{x}}_{t_1} + \sum_{t=t_1}^{t_2-1} \Delta\widetilde{s}_t, \\
\Delta\widetilde{s}_t &= \tilde{\mathbf{x}}_{t+1} - \tilde{\mathbf{x}}_t,
\end{aligned}
\tag{10}
$$

where $\Delta\widetilde{s}_t$ denotes the change of the seasonal pattern, $\Delta\widetilde{s}_t \in \mathbb{R}^T$. The shift reflects the variation between small time steps, but when such variation (shift) builds up over a rather long period, the trend $\mathbf{d}$ comes out. It can be achieved as $\sum_{t=t_1}^{t_2-1} \Delta\widetilde{s}_t$. Therefore, we can model trends by capturing the long- and short-range dependencies of shifts among different time steps.

Next, we introduce a smoothing filter attention mechanism to construct multi-scale transformation layers. A difference attention module is mounted to capture and interconnect shifts of the corresponding scale. These mechanisms make our Diviner capture multi-scale variations in non-stationary time series, and the mathematical description is listed below.

**Diviner input layer**. Given the time series data $\mathbf{X}$, we transform $\mathbf{X}$ into $\widetilde{\mathbf{X}} = \begin{bmatrix} \tilde{\mathbf{x}}_1 & \tilde{\mathbf{x}}_2 & \dots & \tilde{\mathbf{x}}_K \end{bmatrix}$, where $\tilde{\mathbf{x}}_i$ is a vector of length $T$ with the time series data per day (seasonal), and $K$ denotes the number of days, $\tilde{\mathbf{x}}_i \in \mathbb{R}^T, \widetilde{\mathbf{X}} \in \mathbb{R}^{T \times K}$. Then we construct the dual input for Diviner. Noticing that Diviner adopts an encoder-decoder architecture, we construct $\mathbf{X}_{en}^{in}$ for encoder and $\mathbf{X}_{de}^{in}$ for decoder, where $\mathbf{X}_{en}^{in} = \begin{bmatrix} \tilde{\mathbf{x}}_1 & \tilde{\mathbf{x}}_2 & \dots & \tilde{\mathbf{x}}_K \end{bmatrix}, \mathbf{X}_{de}^{in} = \begin{bmatrix} \tilde{\mathbf{x}}_{K-K_{de}+1} & \tilde{\mathbf{x}}_{K-K_{de}} & \dots & \tilde{\mathbf{x}}_K \end{bmatrix}$, and $\mathbf{X}_{en}^{in} \in \mathbb{R}^K$, $\mathbf{X}_{de}^{in} \in \mathbb{R}^{K_{de}}$. This means that $\mathbf{X}_{en}^{in}$ takes all elements from $\widetilde{\mathbf{X}}$ while $\mathbf{X}_{de}^{in}$ takes only the latest $K_{de}$ elements. After that, a fully connected layer on $\mathbf{X}_{en}^{in}$ and $\mathbf{X}_{de}^{in}$ is used to obtain $\mathbf{E}_{en}^{in}$ and $\mathbf{E}_{de}^{in}$, where $\mathbf{E}_{en}^{in} \in \mathbb{R}^{d_m \times K}$, $\mathbf{E}_{de}^{in} \in \mathbb{R}^{d_m \times K_{de}}$ and $d_m$ denotes the model dimension.

**Smoothing filter attention mechanism**. Inspired by Nadaraya-Watson regression[51,52] bringing the adjacent points closer together, we introduce the smoothing filter attention mechanism with a learnable kernel function and self-masked architecture, where the former brings similar items closer to filter out the random component and adjust the non-stationary data to stable features, and the letter reduces outliers. The smoothing filter attention mechanism is implemented based on the input $\mathbf{E} = \begin{bmatrix} \boldsymbol{\xi}_1 & \boldsymbol{\xi}_2 & \dots & \boldsymbol{\xi}_{K_{in}} \end{bmatrix}$, where $\boldsymbol{\xi}_i \in \mathbb{R}^{d_m}$, $\mathbf{E}$ is the general reference to the input of each layer, for encoder $K_{in} = K$, and for decoder $K_{in} = K_{de}$. Specifically, $\mathbf{E}_{en}^{in}$ and $\mathbf{E}_{de}^{in}$ are, respectively, the input of the first encoder and decoder layer. The calculation process is shown as follows:

$$
\boldsymbol{\eta}_i = \frac{\sum_{j \neq i} K(\boldsymbol{\xi}_i, \boldsymbol{\xi}_j) \odot \boldsymbol{\xi}_j}{\sum_{j \neq i} K(\boldsymbol{\xi}_i, \boldsymbol{\xi}_j)},
\tag{11}
$$

$$
K(\boldsymbol{\xi}_i, \boldsymbol{\xi}_j) = exp(\mathbf{w}_i \odot (\boldsymbol{\xi}_i - \boldsymbol{\xi}_j)^2),
\tag{12}
$$

where $\mathbf{w}_i \in \mathbb{R}^{d_m}, i \in [1, K_{in}]$ denotes the learnable parameters, $\odot$ denotes the element-wise multiple, $(\cdot)^2$ denotes the element-wise square and the square of a vector here represents the element-wise square. To simplify the representation, we assign the smoothing filter attention mechanism as Smoothing-Filter($\mathbf{E}$) and denote its output as $\mathbf{H}_s$. Before introducing our difference attention module, we first define the difference between a matrix and its inverse operation CumSum.

**Difference and CumSum operation**. Given a matrix $\mathbf{M} \in \mathbb{R}^{m \times n}$, $\mathbf{M} = \begin{bmatrix} \mathbf{m}_1 & \mathbf{m}_2 & \dots & \mathbf{m}_n \end{bmatrix}$, the difference of $\mathbf{M}$ is defined as:

$$
\Delta\mathbf{M} = \begin{bmatrix} \Delta\mathbf{m}_1 & \Delta\mathbf{m}_2 & \dots & \Delta\mathbf{m}_n \end{bmatrix},
\tag{13}
$$

where $\Delta\mathbf{m}_i = \mathbf{m}_{i+1} - \mathbf{m}_i, \Delta\mathbf{m}_i \in \mathbb{R}^m, i \in [1, n)$ and we pad $\Delta\mathbf{m}_n$ with $\Delta\mathbf{m}_{n-1}$ to keep a fixed length before and after the difference operation. The CumSum operation $\Sigma$ toward $M$ is defined as:

$$
\Sigma\mathbf{M} = \begin{bmatrix} \Sigma\mathbf{m}_1 & \Sigma\mathbf{m}_2 & \dots & \Sigma\mathbf{m}_n \end{bmatrix},
\tag{14}
$$

where $\Sigma\mathbf{m}_i = \sum_{j=1}^{i} \mathbf{m}_j, \Sigma\mathbf{m}_i \in \mathbb{R}^m$. The differential attention module, intuitively, can be seen as an attention mechanism plugged between these two operations, mathematically described as follows.

**Differential attention module**. The input of this model involves three elements: $\mathbf{Q}, \mathbf{K}, \mathbf{V}$. The $(\mathbf{Q}, \mathbf{K}, \mathbf{V})$ varies between the encoder and decoder, which is $(\mathbf{H}_s^{en}, \mathbf{H}_s^{en}, \mathbf{H}_s^{en})$ for the encoder and $(\mathbf{H}_s^{de}, \mathbf{E}_{en}^{out}, \mathbf{E}_{en}^{out})$ for the decoder, where $\mathbf{E}_{en}^{out}$ is the embedded result of the final encoder block (assigned in the pseudo-code), $\mathbf{H}_s^{en} \in \mathbb{R}^{d_m \times K}, \mathbf{H}_s^{de} \in \mathbb{R}^{d_m \times K_{de}}, \mathbf{E}_{en}^{out} \in \mathbb{R}^{d_m \times K}$.

$$\mathbf{Q}_s^{(i)}, \mathbf{K}_s^{(i)}, \mathbf{V}_s^{(i)} = \mathbf{W}_q^{(i)}\Delta\mathbf{Q} + \mathbf{b}_q^{(i)}, \mathbf{W}_k^{(i)}\Delta\mathbf{K} + \mathbf{b}_k^{(i)}, \mathbf{W}_v^{(i)}\Delta\mathbf{V} + \mathbf{b}_v^{(i)}, \quad (15)$$

$$\widetilde{\mathbf{V}}_s^{(i)} = \mathbf{V}_s^{(i)} \cdot \text{SoftMax}\left(\frac{\mathbf{Q}_s^{(i)\top} \cdot \mathbf{K}_s^{(i)}}{\sqrt{d_m}}\right), \quad (16)$$

$$\mathbf{D} = \Sigma(\mathbf{W}_s\left[\begin{array}{cccc}\widetilde{\mathbf{V}}_s^{(1)\top} & \widetilde{\mathbf{V}}_s^{(2)\top} & \ldots & \widetilde{\mathbf{V}}_s^{(h)\top}\end{array}\right]^\top), \quad (17)$$

where $\mathbf{W}_q^{(i)} \in \mathbb{R}^{d_a \times d_m}, \mathbf{W}_k^{(i)} \in \mathbb{R}^{d_{attn} \times d_m}, \mathbf{W}_v^{(i)} \in \mathbb{R}^{d_a \times d_m}, \mathbf{W}_s \in \mathbb{R}^{d_m \times hd_a}$, $\mathbf{D} \in \mathbb{R}^{d_m \times K}, i \in [1, h], h$ denotes the number of parallel attentions. $\left[\cdot\right]$ denotes the concatenation of matrix, $\widetilde{\mathbf{V}}_s^{(i)}$ denotes the deep shift, and $\mathbf{D}$ denotes the deep trend. We denote the differential attention module as Differential-attention$(\mathbf{Q}, \mathbf{K}, \mathbf{V})$ to ease representations.

**Convolution Generator**. The final output of Diviner is calculated through convolutional layers, called the one-step generator, which takes the output of the final decoder layer $\mathbf{E}_{de}^{out}$ as the input:

$$\mathbf{R}_{predict} = \text{ConvNet}(\mathbf{E}_{de}^{out}), \quad (18)$$

where $\mathbf{R}_{predict} \in \mathbb{R}^{d_m \times K_r}, \mathbf{E}_{de}^{(M)} \in \mathbb{R}^{d_m \times K_{de}}$, ConvNet is a multilayer fully convolution net, whose input and output channels are the input length of the decoder $K_{de}$ and the prediction length $K_r$, respectively.

**Pseudo-code of Diviner**. For the convenience of reproducing, We summarize the framework of our Diviner in the following pseudo-code:

---

**Algorithm 1** Pseudo-code for Diviner Architecture.

---

**Input:** Encoder input time series data $\mathbf{X}_{en}^{in}$, decoder input time series data $\mathbf{X}_{de}^{in}$, encoder layers number $N$, decoder layers number $M$, generator layers number $L$, where $\mathbf{X}_{en}^{in} \in \mathbb{R}^{T \times K}, \mathbf{X}_{de}^{in} \in \mathbb{R}^{T \times K_{de}}$.
**Output:** Diviner prediction result $R_{predict}$.
#initialization
$\mathbf{E}_{en}^{(0)} = \text{Input-Layer}(\mathbf{X}_{en}^{in})$ ▷ $\mathbf{E}_{en}^{(0)} \in \mathbb{R}^{d_m \times K}$
$\mathbf{E}_{de}^{(0)} = \text{Input-Layer}(\mathbf{X}_{de}^{in})$ ▷ $\mathbf{E}_{de}^{(0)} \in \mathbb{R}^{d_m \times K}$
#Diviner Encoder Layer
**for** $l$ **in** $\{1, ...N\}$ **do**
 $\mathbf{H}_s^{en,(l)} = \text{Smoothing-Filter}(\mathbf{E}_{en}^{(l-1)})$ ▷ $\mathbf{H}_s^{en,(l)} \in \mathbb{R}^{d_m \times K}$
 $\mathbf{D}_{en}^{(l)} = \text{Differential-attention}(\mathbf{H}_s^{en,(l)}, \mathbf{H}_s^{en,(l)}, \mathbf{H}_s^{en,(l)})$ ▷ $\mathbf{D}_{en}^{(l)} \in \mathbb{R}^{d_m \times K_{de}}$
 $\mathbf{E}_{en}^{(l)} = \text{Linear}(\mathbf{D}_{en}^{(l)}) + \mathbf{D}_{en}^{(l)} + \mathbf{H}_s^{en,(l)}$ ▷ $\mathbf{E}_{en}^{(l)} \in \mathbb{R}^{d_m \times K_{de}}$
**end**
$\mathbf{E}_{en}^{out} = \mathbf{E}_{en}^{(N)}$ ▷ $\mathbf{E}_{en}^{(l)} \in \mathbb{R}^{d_m \times K_{de}}$
#Diviner Decoder Layer
**for** $l$ **in** $\{1, ...M\}$ **do**
 $\mathbf{H}_s^{de,(l)} = \text{Smoothing-Filter}(\mathbf{E}_{de}^{(l-1)})$ ▷ $\mathbf{H}_s^{de,(l)} \in \mathbb{R}^{d_m \times K_{de}}$
 $\mathbf{D}_{de}^{(l)} = \text{Differential-attention}(\mathbf{H}_s^{de,(l)}, \mathbf{E}_{en}^{out}, \mathbf{E}_{en}^{out})$ ▷ $\mathbf{D}_{de}^{(l)} \in \mathbb{R}^{d_m \times K_{de}}$
 $\mathbf{E}_{de}^{(l)} = \text{Linear}(\mathbf{D}_{de}^{(l)}) + \mathbf{D}_{de}^{(l)} + \mathbf{H}_s^{de,(l)}$ ▷ $\mathbf{E}_{de}^{(l)} \in \mathbb{R}^{d_m \times K_{de}}$
**end**
#Diviner Generator Layer
$\mathbf{E}_{de}^{out} = \mathbf{E}_{de}^{(M)}$ ▷ $\mathbf{E}_{de}^{out} \in \mathbb{R}^{d_m \times K_{de}}$
$\mathbf{E}_{ge}^{(0)} = \text{1D-Conv}(\mathbf{E}_{de}^{out})$ ▷ $\mathbf{E}_{ge}^{(0)} \in \mathbb{R}^{d_m \times K_r}$
 **for** $l$ **in** $\{1, ...L-1\}$ **do**
 $\mathbf{E}_{ge}^{(l)} = \text{1D-Conv}(\mathbf{E}_{ge}^{(l-1)}) + \mathbf{E}_{ge}^{(l-1)}$ ▷ $\mathbf{E}_{ge}^{(l)} \in \mathbb{R}^{d_m \times K_r}$
 **end**
$\mathbf{R}_{predict} = \mathbf{E}_{ge}^{(L)}$ ▷ $\mathbf{R}_{predict} \in \mathbb{R}^{d_m \times K_r}$
 **return** $\mathbf{R}_{predict}$

---

## Data availability

The datasets supporting our work have been deposited at https://doi.org/10.5281/zenodo.7827077. However, restrictions apply to the availability of NPT data, which were used under license for the current study, and so are not publicly available. Data are, however, available from the authors upon reasonable request and with permission of China Information Technology Designing Consulting Institute.

## Code availability

Codes are available at https://doi.org/10.5281/zenodo.7825740.

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

## Acknowledgements

This work was supported by the National Natural Science Foundation of China under Grant 62076016 and 12201024, Beijing Natural Science Foundation L223024.

## Author contributions

Y.Y., S.G., B.Z., J.Z., and D.D. conceived the research. All authors work on the writing of the article. Y.Y. and S.G. equally contributed to this work by performing experiments and results analysis. Z.W. and Y.Z. collected the 5G network traffic data. All authors read and approved the final paper.

## Competing interests

The authors declare no competing interests.

## Inclusion and ethics

No 'ethics dumping' and 'helicopter research' cases occurred in our research.
