## [Peer Review file · Communications Engineering]

Long-term 5G network traffic forecasting via modeling non-stationarity with deep learningReviewers' comments:

Reviewer #1 (Remarks to the Author):

In this paper, The authros used deep learning models for forecasting 5G network traffic. The paper is a kind of application paper. The paper can be accepted with minor revisions.

- 1-The authors should increase the number of benchmarks in the application.
- 2- Please, review and cite the recent literature about forecasting with deep leraning
- 3- The MASE metric should use as an error metric in the applications.

Reviewer #2 (Remarks to the Author):

1. This paper is not technically correct.
2. The author have not clearly expressed the idea and novelty in the paper.
3. The references are not sufficient.
4. The work is not novel and not relevant to the journal.
5. The length of the paper is not sufficient.
6. Some of the work is repetitive, so it is advised to give proper justification about it.
7. The title and abstract is not written properly
8. The quality of diagrams should be improved and should be of 600dpi.
9. Conclusion should be explained in more detail

Reviewer #3 (Remarks to the Author):

The paper addresses a relevant matter in the context of 5G networks. Despite there is wide SOTA in this research area, the authors provide sufficient motivation and novelty in their proposed solution. Results are sound, exploiting real datasets and expanding the assessment of the solution to other domains. Overall, in this reviewer's perspective, it is a good paper.

Despite the paper in its current form is easy-to-follow, the structure is somehow unconventional.....introduction includes partly description of solution and anticipation of results.....the detailed description of the proposed approach is provided after results....Authors should probably re-organize the sections, while keeping the contents.

The Response Letter for COMMS-22-0216-T

We sincerely thank reviewers for their valuable suggestions and the support of the novelty and results of our work: R1: "can be accepted with minor revisions", R3: "Results are sound", "sufficient motivation and novelty", "a good paper". We believe that all comments have been carefully accommodated to the best of our knowledge. The underlined texts in the response letter have been highlighted in blue in the current manuscript, and the point-by-point responses to all comments are as follows.

Reviewer: #1

Summary

In this paper, the authors used deep learning models for forecasting 5G network traffic. The paper is a kind of application paper. The paper can be accepted with minor revisions

Comment-1.1: The authors should increase the number of benchmarks in the application.

Thanks for your suggestion. We provide more benchmarks to validate the effectiveness of the proposed method over other deep learning baselines. Table 1+, Table 2+, Table 3+ below exhibit the supplemental experiment results.

1. NPT-2 & NPT-3 datasets (Line 254-264, Line 312-321): We collect another two groups of network traffic data (NPT-2, NPT-3), where each group includes five network ports data recorded every 15 minutes for one year and *inbits* is the target feature. The prediction horizon follows the setting in the original NPT dataset (named NPT-1 in the current version), where we forecast the entire network traffic series {1, 3, 7, 14, 30} days ahead (aligned with {96, 288, 672, 1344, 2880} prediction spans in Table 1+), and the capacity utilization {30, 60, 90} days ahead (aligned with {2880, 5760, 8640} prediction spans ahead in Table 2+). The supplemental datasets reproduce the experimental results on the original network traffic data (NPT-1), where

the proposed method can not only perform a precise long-term network traffic prediction for up to three months, but also surpasses current arts involving accuracy and robustness by a large margin. In addition, we have the following results by sorting the comprehensive performance (obtained by the average MASE errors) of the baselines established with the Transformer framework: Diviner > Autoformer > Transformer > Informer. This result verifies our proposal that incorporating the non-stationarity promotes the adaptive ability of neural networks to model time series ({Diviner, Autoformer}>{Transformer, Informer}), and the modeling of multi-scale non-stationarity other breaks through the ceiling of prediction abilities for deep learning models. Furthermore, considering diverse network traffic patterns in the provided datasets (about 50 ports), these experiments prove the generalization of the proposed Diviner to deal with a wide range of non-stationary time series.

2. Solar energy production prediction (Line 394-411). The solar dataset contains the 10-minute level one-year (2006) solar power production data of 137 PV plants in Alabama State, and *PV-136* is the target feature. Given that the amount of solar energy produced daily is generally stable, conducting a super long-term prediction is unnecessary. Therefore, we set the prediction horizon to {1, 2, 5, 6} days, aligned with {144, 288, 720, 864} prediction steps ahead. Furthermore, this characteristic of solar energy means that its production series tend to be stationary, and thereby the comparison of the predictive performances between different models on this dataset presents their basic series modeling abilities. Concretely, considering the MASE error can be used to assess model's performances on different series, we calculate and sort each model's average MASE error under different prediction horizons to measure their time series modeling abilities. The results are as follows: Diviner > NBeats > Transformer > Autoformer > Informer > LSTM, where Diviner surpasses all Transformer-based models in the selected baselines. Provided that the series data is not that non-stationary, the advantages of Autoformer's modeling time series non-stationarity are not apparent. At the same time, capturing stable long- and short-term dependencies is still effective.

3. Road occupancy rate prediction (Line 412-433). The Traffic dataset contains hourly

two-year (2015~2016) road occupancy rate collected from 862 sensors on San Francisco Bay area freeways by the California Department of Transportation, where *sensor861* is the target feature. The prediction horizon is set to {7, 14, 30, 40} days, aligned with {168, 336, 720, 960} prediction steps ahead. Considering the road occupancy rate tends to have a weekly cycle, we use this dataset to compare different networks' ability to model the temporal cycle. During the comparison, we mainly focus on the following two groups of models: group-1 takes the non-stationary specialization of time series into account (Diviner, Autoformer), and group-2 does not employ any time-series-specific components (Transformer, Informer, LSTMa). We find that group-1 gains a significant performance improvement over group-2, which suggests the necessity of modeling non-stationarity. As for the proposed Diviner model, it achieves a 27.64% MAE reduction (0.604 → 0.437) to the Transformer model when it is used to forecast 30-day road occupancy rate. Subsequently, we conduct an intra-group comparison for group-1, where the Diviner still gains an average 35.37% MAE reduction (0.523 → 0.338) to Autoformer. We attribute this to Diviner's multiple-scale modeling of non-stationarity, while the trend-seasonal decomposition of Autoformer merely reflects time series variation at the particular scale. These experimental results demonstrate that Diviner is competent in predicting time series data with cycles.

Thanks again for your valuable advice. The supplemental experiments demonstrate the desirable advantages of our approach. We have revised the corresponding parts of the manuscript, and highlighted the modified parts in blue.

Comment-1.2: Please, review and cite the recent literature about forecasting with deep learning.

Reply-1.2: Thanks for your suggestion. We investigate the recent international conferences and journals, and the literature about forecasting with deep learning of 2021~2022 has been sorted out in the *supplement reference*.

Following the introduction to existing approaches in our manuscript (Line 54-88), the RNN-based models mentioned in [28, 29, 30, 31, 32, 33, 34] features a feedback loop

that allows models to memorize historical data and process variable-length sequences as both inputs and outputs. [35, 36, 27] develop their model on Transformer framework to capture longer dependencies and interactions within series data, [42, 43, 44, 45, 46, 47] attempt to incorporate time series decomposition into deep learning models to increase model's robustness when modeling non-stationary time series.

The supplement references have been added to our manuscript.

Comment-1.3: The MASE metric should use as an error metric in the applications.

Reply-1.3: Following your suggestion, we add MASE as an error metric in all benchmarks, the experimental results are provided as follows, and complete experimental results have been updated in our manuscript.

Table 1+: Time-series forecasting results on the 5G traffic network dataset

Metric (MASE)		Diviner	Auto-former	In-former	Trans-former	ARIMA	Prophet	NBeats	LSTMa
NPT-1	96	1.391	2.090	1.427	1.362	1.992	2.172	2.082	4.468
	288	1.598	2.104	2.488	1.876	2.328	2.180	2.927	4.698
	672	1.601	2.278	3.862	1.907	2.941	2.269	2.684	4.814
	1344	1.585	2.017	3.987	1.996	3.949	2.298	2.989	4.817
	2880	1.613	2.601	4.343	2.697	6.750	2.382	3.491	5.050
NPT-2	96	1.800	2.681	2.870	2.088	2.208	2.005	2.116	4.135
	288	1.977	2.786	3.788	2.082	2.289	2.269	2.481	4.437
	672	2.074	2.357	3.864	2.453	2.237	2.306	2.685	4.280
	1344	1.814	2.301	4.181	2.889	2.453	2.496	2.593	4.666
	2880	1.861	3.072	4.457	2.240	2.375	2.634	3.459	4.794
NPT-3	96	1.672	2.076	3.397	2.150	2.476	2.457	1.797	4.834
	288	1.558	3.144	3.736	2.478	2.487	2.386	2.579	4.679
	672	1.599	2.753	4.044	2.147	2.529	2.457	2.558	4.969
	1344	1.822	2.569	4.520	2.363	2.657	2.558	3.128	5.050
	2880	1.756	3.228	4.622	2.177	2.416	2.579	3.374	5.137

Table 2+: Long-term capacity utilization forecasting results on NPT dataset

Models		Diviner		Baseline-A		Baseline-M	
Metric (MASE)		Inbits	Outbits	Inbits	Outbits	Inbits	Outbits
NPT-1	96	1.947	3.420	4.851	5.415	17.690	35.274
	288	4.352	6.494	5.371	6.848	194.431	127.160
	672	3.425	4.172	4.693	6.084	113.567	87.096
	1344	1.992	2.574	5.159	4.442	14.959	10.341
	2880	2.817	2.905	5.751	4.980	46.936	26.470
NPT-2	96	4.363	3.113	6.265	5.766	44.602	34.165
	288	2.232	2.987	5.291	5.192	27.947	37.698
	672	4.713	4.911	6.425	5.580	42.245	102.973
	1344	3.611	4.062	5.622	5.567	30.023	198.393
	2880	1.947	3.420	4.851	5.415	17.690	35.274
NPT-3	96	4.352	6.494	5.371	6.848	194.431	127.160
	288	3.425	4.172	4.693	6.084	113.567	87.096
	672	1.992	2.574	5.159	4.442	14.959	10.341
	1344	2.817	2.905	5.751	4.980	46.936	26.470
	2880	4.363	3.113	6.265	5.766	44.602	34.165

Table 3+: Time-series forecasting results on other real-world datasets

Metric (MASE)	Diviner	Auto-former	In-former	Trans-former	ARIMA	Prophet	NBeats	LSTMa	
WTH	144	7.072	9.125	8.316	10.038	17.151	24.099	12.236	18.956
	432	8.135	9.235	8.944	9.754	16.872	22.974	10.397	18.470
	1008	6.806	12.720	8.030	10.666	16.185	20.294	10.167	18.364
	2016	6.348	21.140	8.651	8.651	17.426	17.240	13.173	18.671
	4320	5.832	13.897	13.131	10.035	9.042	18.377	7.070	13.783
ETTh1	24	4.174	5.338	5.634	13.664	6.478	6.273	6.136	6.204
	48	4.629	5.222	7.274	11.950	9.669	7.525	6.659	8.164
	168	5.977	6.387	7.893	14.099	11.498	17.407	12.228	8.943
	336	6.031	7.742	8.710	9.385	13.346	40.962	11.118	15.709
	720	7.195	8.133	9.720	9.943	17.116	72.690	18.859	17.161
ETTh2	24	4.281	5.927	5.062	13.773	9.386	8.036	6.707	6.475
	48	5.209	5.871	6.491	15.298	9.799	9.551	8.104	7.194
	168	6.070	6.881	6.708	9.105	9.743	18.418	9.450	8.864
	336	5.792	6.258	6.475	7.158	11.460	39.489	10.171	9.410
	720	5.435	9.172	6.365	6.247	15.420	68.888	10.073	10.058
ETTm1	24	4.710	12.952	6.452	19.687	9.702	13.659	64.998	10.974
	48	5.159	10.838	9.971	20.853	13.266	13.223	52.892	17.818
	96	6.957	8.503	15.975	19.325	17.135	17.006	47.497	18.036
	288	9.375	10.990	24.850	18.391	25.030	25.748	27.856	26.196
	672	9.323	12.195	28.456	23.861	30.798	51.876	21.210	38.575
ECL	168	2.315	2.937	3.225	3.597	5.598	8.163	2.327	4.200
	336	2.602	3.267	3.478	4.215	5.770	20.269	2.364	5.915
	720	2.544	5.643	3.552	3.278	5.804	8.803	2.998	6.010
	960	3.849	7.242	4.786	4.699	7.730	33.566	4.250	7.919
Ex-change	10	2.867	3.203	21.601	25.964	5.227	10.922	7.129	45.551
	20	3.160	4.054	18.341	18.949	5.810	8.303	7.049	34.092
	30	3.132	4.945	13.945	16.310	5.378	7.037	6.839	28.634
	60	4.265	4.948	19.557	18.109	8.653	8.264	10.648	30.365
Solar	144	7.461	11.091	8.290	11.742	31.093	29.165	8.487	18.917
	288	8.355	12.035	10.007	9.289	34.362	32.236	8.988	20.327
	720	8.793	13.497	13.803	11.352	35.099	32.930	9.176	21.886
	864	7.053	14.423	21.299	11.367	32.419	30.415	8.488	20.030
Traffic	168	0.835	1.561	3.729	2.071	2.953	2.957	1.700	3.866
	336	0.847	1.548	3.738	1.857	2.977	2.977	1.714	3.934
	720	1.457	1.751	3.836	2.014	3.069	3.067	1.779	4.087
	960	1.299	1.986	3.809	3.649	3.017	3.013	1.740	3.938

Thanks again for your advice. The MASE metric enables us to calculate the model's comprehensive performance with experimental results under different prediction horizon settings, which has been employed in the analysis of NPT-2 & NPT-3 datasets and solar energy production prediction (in Comment-1.1).

Reviewer: #2

Comment-2.1: This paper is not technically correct.

We address your concerns from the following aspects below:

1. *In terms of model design*, we develop our model to solve the challenges brought by non-stationarity: (a) Multi-scale temporal variation. (b) Random factors. (c) Distribution shift. We introduce a deep stationary process to tackle these challenges to model the non-stationary time series with multi-scale stable features. Concretely, we present a smoothing filter attention mechanism to filter out random components (*challenge-b*) and adjust the feature scale of the time series layer-by-layer (*challenge-a*). Simultaneously, a difference attention module is designed to calculate long- and short-range dependencies by capturing the stable shifts at the corresponding scale (*challenge-c*). Detailed descriptions are stated in Page 4~5 in our original submission.
2. *In terms of effectiveness*, we collect three groups (49 ports) of network port traffic datasets (NPT) from real-world metropolitan networks that deliver 5G services by China Unicom to demonstrate the proposed methods' effectiveness. We also compare the proposed method with numerous prior arts over multiple applications involving climate, control, electricity, economics, energy, and transportation fields to validate the effectiveness of our method. The detailed descriptions are stated in Page 7~11 in the original submission.
3. *In terms of technical soundness*, we incorporate stationary processes into a well-designed hierarchical structure to model non-stationary time series with multi-scale stable features, where stochastic process theories guarantee the prediction of stationary events [48, 49, 50]. Besides, our experiments demonstrate that the proposed Diviner, as a deep learning model, can deal with a wide range of non-stationary time series, providing strong evidences on our technical correctness.

Comment-2.2: The author have not clearly expressed the idea and novelty in the paper.

To make our idea easier to follow, we make the following adjustment:

(a) We further clarify the motivation of our idea more clearly (Line 34-41):

Existing methods for 5G network traffic forecasting suffer a severe performance degeneration since the long-term prediction horizon exposes the non-stationarity of time series. This inherent non-stationarity of real-world time series data is caused by multi-scale temporal variations, random perturbations and outliers, which present various challenges: 1. Multi-scale temporal variations. 2. Random factors. 3. Distribution shift. To address these issues, we incorporate stationary processes into a well-designed hierarchical structure and model non-stationary time series with multi-scale stable features. By doing so, we achieve a robust and precise long-term prediction on real-world 5G time series forecasting data as shown in extensive experiments in Page 8-10 in our original submission.

(b) We elaborate more on existing methods' shortcomings when solving non-stationarity issues (Line 54-88):

1. RNN+: RNN-based models [28, 29, 30, 31, 32, 33, 34] feature a feedback loop that allows models to memorize historical data, which calculates the cumulative dependency between time steps. Nevertheless, such indirect modeling of temporal dependencies can not disentangle information from different scales within historical data and thus fails to capture multi-scale variations within non-stationary time series.

2. Transformer+: Transformer-based models [35, 36, 37] use global self-attention mechanism to capture the long-term dependencies within time series. However, despite their promising long-term forecasting results, time series' specialization is not taken into account during their modeling process, where varying distributions of non-stationary time series deteriorate their predictive performances.

3. Recent researches incorporating time series decomposition+: Recent works [42, 43, 44, 45, 46, 47] attempt to incorporate time series decomposition into deep learning models. Their limited decomposition, e.g., trend-seasonal decomposition, reverses the correlation between components and merely presents the variation of time series at particular scales.

To make our novelty easier to follow, we summarize our novelty in the concluding part of the Introduction (Line 98-111):

1. We explore a new avenue to solve the challenges presented in long-term time series prediction by modeling non-stationarity in the deep learning paradigm. This line is much more universal and effective than the previous incorporating temporal decomposition for its limited decomposition that merely presents the temporal variation at particular scales
2. We develop a new framework with a well-designed hierarchical structure to model the multi-scale stable regularities within non-stationary time series. In contrast to previous methods employing various modules in the same layer, we perform a dynamical scale transformation between different layers and model stable temporal dependencies in the corresponding layer. This hierarchical deep stationary process synchronizes with the cascading feature embedding of deep neural networks, which enables us to capture complex regularities contained in the long-term histories and achieves precise long-term network traffic forecasting.

The above contents have been added to the manuscript correspondingly, and we would sincerely appreciate it if the above supplements are helpful and address your concern.

Comment-2.3: The references are not sufficient.

Thanks for your suggestion. We review the recent literature about employing network traffic prediction in practical applications and also the recent deep learning forecasting methods. The additional references are sorted out in the *supplement reference*.

1. *The literature employing network traffic prediction in applications* (Line 436-445). Although some literature [62, 63, 64] in the early stage argues that the probabilistic traffic forecast under uncertainty is more suitable for the varying network traffic rather than a concrete forecast produced by time series models, the probabilistic traffic forecast and the concrete traffic forecast share the same historical information in essence. And the development of time series forecasting techniques these years has witnessed a series of works employing time series forecast techniques for practical applications such as bandwidth management [14, 15], resource allocation [16], resource provisioning [17].

2. *The literature forecasts with deep learning* (Line 54-88). Following our manuscript's introduction to existing approaches, the RNN-based models mentioned in [28, 29, 30, 31, 32, 33, 34] feature a feedback loop that allows models to memorize historical data and process variable-length sequences as both inputs and outputs. [35, 36, 37] develop their model on Transformer framework to capture longer dependencies and interactions within series data, [42, 43, 44, 45, 46, 47] attempt to incorporate time series decomposition into deep learning models to increase model's stability when modeling non-stationary time series.

The supplement references have been added to our manuscript.

Comment-2.4: The work is not novel and not relevant to the journal.

In terms of novelty, a similar concern is addressed in Comment-2.1, w.r.t. the following content (Line 98-111):

1. We explore a new avenue to solve the challenges presented in long-term time series prediction by modeling non-stationarity in the deep learning paradigm. This line is much more universal and effective than the previous incorporating temporal decomposition for its limited decomposition that merely presents the temporal variation at particular scales.

2. We develop a new framework with a well-designed hierarchical structure to model the multi-scale stable regularities within non-stationary time series. In contrast to previous methods employing various modules in the same layer, we perform a dynamical scale transformation between different layers and model stable temporal dependencies in the corresponding layer. This hierarchical deep stationary process synchronizes with the cascading feature embedding of deep neural networks, which enables us to capture complex regularities contained in the long-term histories and achieves precise long-term network traffic forecasting.

In terms of whether this work is relevant to the journal, we had carefully read the Aims & Scope of the Communication Engineering journal and sorted out the following statement to support our work published in Communication Engineering journal:

(a) Scope: We are also interested in submissions at the intersection of engineering with other scientific disciplines, such as biology, chemistry, physics, where the central advance of the study is of interest to other engineers.

(b) Aims: In general, to be acceptable, a paper should report new thinking with resulting advances in opportunity, capability or benefit.

We regard our work as an intersection of communication engineering with artificial intelligence (covered in scope), which aims to solve the crucial bottleneck in making a detailed long-term network traffic forecasts for massive network ports with deep learning techniques (satisfying journal aims). Therefore, this work suits the topic and criteria of the *Communication Engineering* journal. We would sincerely appreciate it if the above statements are helpful and address your concern.

Comment-2.5: The length of the paper is not sufficient.

Thanks for your suggestion, we add the following content:

1. We add several 5G network traffic benchmarks and apply our method to more engineering fields such as solar energy production forecasting and road occupancy forecasting, *w.r.t.* Line 394-433 in our manuscript.

2. We reevaluate the performance of baseline models on each benchmark with MASE as a new indicator, where we can assess the model's comprehensive predictive performance by calculating average MASE errors under different prediction steps. Subsequently, we perform a more detailed analysis toward the necessity of modeling non-stationarity with our experimental results, *w.r.t.* Line 254-264, Line 312-321, Line 394-411 in our manuscript.

3. We discuss the effectiveness of using time series prediction methods to assist network planning, and review more literature to prove this, *w.r.t.* Line 436-445 in our manuscript.

4. We discuss the shortcomings of existing methods when solving the non-stationarity issue *w.r.t.* Line 52-88 in our manuscript.

We have added the supplemental contents to our manuscript correspondingly.

Comment-2.6: Some of the work is repetitive, so it is advised to give proper justification about it.

Thanks for your suggestion. We make the following adjustments to solve this problem:

1. We delete the description of the solution and results in the Introduction, which has already been shown in Section 2.1 and Section 2.2, respectively. Instead, we add a clear conclusion of our work (see Comment-2.9).
2. We delete the background of 5G network construction in the Discussion, which has already been elaborated in the Introduction. Instead, we discuss the effectiveness of employing network traffic prediction in practical applications (see Comment-2.3).
3. We delete the description of the experimental results concerning 5G network traffic prediction and electricity transformer temperature prediction in the Discussion.

Comment-2.7: The title and abstract is not written properly

Thanks for your suggestion. We review the *style and formatting guide* of the *Communication Engineering* journal.

In terms of the *title*, we highlight the main innovation of our work by *modeling non-stationarity with deep learning for long-term 5G network traffic forecasting*. Finally, the title is revised to Long-term 5G network traffic forecasting via modeling non-stationarity with deep learning.

In terms of the *abstract*, following the *instruction and the formatting guide*, we incrementally introduce the necessity and motivation of the proposed techniques and clarify the major results of our work. Finally, the abstract is revised to:

5G cellular networks have recently fostered a wide range of emerging applications, but their popularity has led to traffic growth that far outpaces network expansion. This mismatch may decrease network quality and cause severe performance problems. To reduce the risk, operators need long-term traffic prediction to perform network expansion schemes months ahead. However, the long-term prediction horizon exposes the non-stationarity of series data, which deteriorates the performance of existing approaches. We deal with this challenging problem by developing a deep learning model, Diviner, that incorporates stationary processes into a well-designed

hierarchical structure and models non-stationary time series with multi-scale stable features. We demonstrate that Diviner surpasses current arts in 5G network traffic forecasting with substantial performance improvement, enabling detailed months-level forecasting for massive ports with complex flow patterns. Extensive experiments further present its applicability to various predictive scenarios without any modification, showing valuable potential to address broader engineering problems.

The revised abstract and the title name have been updated in our manuscript.

Comment-2.8: The quality of diagrams should be improved and should be of 600dpi.

Thanks for your suggestion. We have converted our diagrams into vector format (.pdf), which can solve the resolution problem in our original .png diagram.

Comment-2.9: Conclusion should be explained in more detail.

Thanks for your suggestion. We rewrite the concluding part in the Introduction and explain our conclusion in more details (Line 89-121):

In this work, we incorporate deep stationary processes into neural networks to achieve precise long-term 5G network traffic forecasts, where stochastic process theories guarantee the prediction of stationary events [48, 49, 50]. Specifically, we develop a deep learning model, Diviner, to discover the multi-scale stable regularities within non-stationary time series and generate precise long-term forecasts. To validate the effectiveness, we collect an extensive network port traffic dataset (NPT) from the intelligent metropolitan network delivering 5G services of China Unicom and compare the proposed model with numerous current arts over multiple applications. We make two distinct research contributions to time series forecasting: (1) We explore a new avenue to solve the challenges presented in long-term time series prediction by modeling non-stationarity in the deep learning paradigm. This line is much more universal and effective than the previous works incorporating temporal decomposition for their limited decomposition that merely presents the temporal variation at particular scales. (2) We develop a new framework with a well-designed hierarchical structure to model the multi-scale stable regularities within non-stationary time series.

In contrast to previous methods employing various modules in the same layer, we perform a dynamical scale transformation between different layers and model stable temporal dependencies in the corresponding layer. This hierarchical deep stationary process synchronizes with the cascading feature embedding of deep neural networks, which enables us to capture complex regularities contained in the long-term histories and achieves precise long-term network traffic forecasting. Our experiment demonstrates that the robustness and predictive accuracy significantly improve as we consider more factors concerning non-stationarity, which provides a new avenue to improve the long-term forecast ability of deep learning methods. Besides, we also show that the modeling of non-stationarity can help discover nonlinear latent regularities within network traffic and achieve a quality long-term 5G network traffic forecast for up to three months. Furthermore, we expand our solution to climate, control, electricity, economic, energy, and transportation fields, which shows the applicability of this solution to multiple predictive scenarios, showing valuable potential to solve broader engineering problems.

We have added the above conclusion to our manuscript.

Reviewer: #3

Summary:

The paper addresses a relevant matter in the context of 5G networks. Despite there is wide SOTA in this research area, the authors provide sufficient motivation and novelty in their proposed solution. Results are sound, exploiting real datasets and expanding the assessment of the solution to other domains. Overall, in this reviewer's perspective, it is a good paper. Despite the paper in its current form is easy-to-follow, the structure is somehow unconventional.....introduction includes partly description of solution and anticipation of results.....the detailed description of the proposed approach is provided after results....Authors should probably re-organize the sections, while keeping the contents.

Response:

We sincerely appreciate your comments that recognize our paper as a good paper providing sufficient motivation, novelty, and sound results.

1. Following your suggestion, we move the original description of the solution and anticipation to the corresponding part in the Results section (the former in Section 2.1 and the latter in Section 2.2). And replace them with a conclusion of our work (Line 89-121):

In this work, we incorporate deep stationary processes into neural networks to achieve precise long-term 5G network traffic forecasts, where stochastic process theories guarantee the prediction of stationary events [48, 49, 50]. Specifically, we develop a deep learning model, Diviner, to discover the multi-scale stable regularities within non-stationary time series and generate precise long-term forecasts. To validate the effectiveness, we collect an extensive network port traffic dataset (NPT) from the intelligent metropolitan network delivering 5G services of China Unicom and compare the proposed model with numerous current arts over multiple applications. We make two distinct research contributions to time series forecasting: (1) We explore a new avenue to solve the challenges presented in long-term time series prediction by modeling non-stationarity in the deep learning paradigm. This line is much more universal and effective than the previous works incorporating temporal decomposition for their limited decomposition that merely presents the temporal variation at particular scales. (2) We develop a new framework with a well-designed hierarchical structure to model the multi-scale stable regularities within non-stationary time series. In contrast to previous methods employing various modules in the same layer, we perform a dynamical scale transformation between different layers and model stable temporal dependencies in the corresponding layer. This hierarchical deep stationary process synchronizes with the cascading feature embedding of deep neural networks, which enables us to capture complex regularities contained in the long-term histories and achieves precise long-term network traffic forecasting. Our experiment demonstrates that the robustness and predictive accuracy significantly improve as we consider more factors concerning non-stationarity, which provides a new avenue to

improve the long-term forecast ability of deep learning methods. Besides, we also show that the modeling of non-stationarity can help discover nonlinear latent regularities within network traffic and achieve a quality long-term 5G network traffic forecast for up to three months. Furthermore, we expand our solution to climate, control, electricity, economic, energy, and transportation fields, which shows the applicability of this solution to multiple predictive scenarios, showing valuable potential to solve broader engineering problems.

We rechecked the *style and formatting guide* and the articles published in the *Communication Engineering* journal, and the Methods section does come after the Results section.

Thanks again for your valuable suggestion. The revised content has been added to the concluding part of current manuscript's Introduction.

Supplement Reference

- [14] Yoo, W. & Sim, A. Time-series forecast modeling on high-bandwidth network measurements. *Journal of Grid Computing* 14 (3), 463–476 (2016)
- [15] Wei, Y., Wang, J. & Wang, C. A traffic prediction based bandwidth management algorithm of a future internet architecture. In *proceedings of International Conference on Intelligent Networks and Intelligent Systems*, 560–563 (2010).
- [16] Garroppo, R. G., Giordano, S., Pagano, M. & Procissi, G. On traffic prediction for resource allocation: A chebyshev bound based allocation scheme. *Computer Communications* 31 (16), 3741–3751 (2008) .
- [17] Bega, D., Gramaglia, M., Fiore, M., Banchs, A. & Costa-Perez, X. Deepcog: Optimizing resource provisioning in network slicing with AI-based capacity forecasting. *IEEE Journal on Selected Areas in Communications* 38 (2), 361–376 (2019).
- [31] Mona, S., Mazin, E., Stefan, L. & Maja, R. Modeling Irregular Time Series with Continuous Recurrent Units. In *Proceedings of International Conference on Machine Learning*, Vol. 162, 19388–19405 (2022).
- [32] Kashif, R., Calvin, S., Ingmar, S. & Roland, V. Autoregressive Denoising Diffusion Models for Multivariate Probabilistic Time Series Forecasting. In *Proceedings of International Conference on Machine Learning*, Vol. 139, 8857–8868 (2021).
- [33] Alasdair, T., Alexander, P. M., Cheng, S. O. & Xie, L. Radflow: A Recurrent, Aggregated, and Decomposable Model for Networks of Time Series. In *Proceedings of International World Wide Web Conference*, 730–742 (2021).
- [34] Ling, F. et al. Multi-task machine learning improves multi-seasonal prediction of the Indian Ocean Dipole. *Nature Communications* 13 (1), 1–9 (2022) .
- [36] Alexandre, D., Étienne, M. & Nicolas, C. TACTiS: Transformer-Attentional Copulas for Time Series. In *Proceedings of International Conference on Machine Learning*, Vol. 162, 5447–5493 (2022).
- [37] Tung, N. & Aditya, G. Transformer Neural Processes: Uncertainty-Aware Meta

Learning Via Sequence Modeling. In Proceedings of International Conference on Machine Learning, Vol. 162, 16569–16594 (2022).

[45] Liu, M. *et al.* SCINet: Time Series Modeling and Forecasting with Sample Convolution and Interaction. In Proceedings of Annual Conference on Neural Information Processing Systems (2022).

[46] Wang, Z. *et al.* Learning Latent Seasonal-Trend Representations for Time Series Forecasting. In Proceedings of Annual Conference on Neural Information Processing Systems (2022).

[47] Xie, C. *et al.* Trend analysis and forecast of daily reported incidence of hand, foot and mouth disease in Hubei, China by Prophet model. Scientific reports 11 (1), 1–8 (2021) .

[62] Geary, N., Antonopoulos, A., Drakopoulos, E., O’Reilly, J. & Mitchell, J. A framework for optical network planning under traffic uncertainty. In the proceedings of International Workshop on Design of Reliable Communication Networks, 50–56 (2001).

[63] Laguna, M. Applying robust optimization to capacity expansion of one location in telecommunications with demand uncertainty. Management Science 44 (11-part-2), S101–S110 (1998).

[64] Bauschert, Thomas *et al.* Network planning under demand uncertainty with robust optimization. IEEE Communications Magazine 52 (2), 178–185 (2014) .

The literature below has already been cited in our first submission but need to be cited again to support the statement in the response letter:

[28] Hochreiter, S. & Schmidhuber, J. Long short-term memory. Neural computation 9 (8), 1735–1780 (1997)

[29] Salinas, D., Flunkert, V., Gasthaus, J. & Januschowski, T. DeepAR: probabilistic forecasting with autoregressive recurrent networks. International Journal of Forecasting 36 (3), 1181–1191 (2020) .

[30] Qin, Y. *et al.* A dual-stage attention-based recurrent neural network for time series prediction. In Proceedings of International Joint Conference on Artificial Intelligence,

2627–2633 (2017).

[35] Vaswani, A. et al. Attention is all you need. In Proceedings of Annual Conference on Neural Information Processing Systems, Vol. 30, 5998–6008 (2017).

[38] Wen, Q. et al. Transformers in time series: A survey. CoRR (2022).
<https://doi.org/10.48550/arXiv.2202.07125>.

[39] Zhou, H. et al. Informer: beyond efficient transformer for long sequence time-series forecasting. In Proceedings of AAAI Conference on Artificial Intelligence (2021).

[40] Kitaev, N., Kaiser, L. & Levskaya, A. Reformer: the efficient transformer. In Proceedings of International Conference on Learning Representations (2019).

[41] Li, S. et al. Enhancing the locality and breaking the memory bottleneck of transformer on time series forecasting. In Proceedings of the 33th Annual Conference on Neural Information Processing Systems, Vol. 32, 5244–5254 (2019).

[42] Wu, H., Xu, J., Wang, J. & Long, M. Autoformer: decomposition transformers with auto-correlation for long-term series forecasting. In Proceedings of Annual Conference on Neural Information Processing Systems, Vol. 34, 22419–22430 (2021).

[43] Zhou, T. et al. Fedformer: frequency enhanced decomposed transformer for long-term series forecasting. In Proceedings of International Conference on Machine Learning, Vol. 162, 27268–27286 (2022).

[44] Liu, S. et al. Pyraformer: low-complexity pyramidal attention for long-range time series modeling and forecasting. In Proceedings of International Conference on Learning Representations (ICLR) (2021).

[48] Cox, D. R. & Miller, H. D. The theory of stochastic processes (Routledge, London, 2017).

[49] Dette, H. & Wu, W. Prediction in locally stationary time series. Journal of Business & Economic Statistics 40 (1), 370–381 (2022) .

[50] Wold, H. O. On prediction in stationary time series. The Annals of Mathematical Statistics 19 (4), 558–567 (1948) .

Reviewers' comments:

Reviewer #1 (Remarks to the Author):

The most important problem of the paper is that MASE metric is not smaller than 1. This means that the random walk is always better than the proposed method. So the paper should be rejected.

Reviewer: #1

Comment-1.1: The most important problem of the paper is that MASE metric is not smaller than 1. This means that the random walk is always better than the proposed method. So the paper should be rejected.

We appreciate your constructive feedback on our paper, and we have carefully considered your comments. We understand that you are concerned about the MASE metric and believe that our proposed method's performance is poor because the MASE value is not smaller than 1. We work more on this issue and we respectfully disagree with your conclusion. The reasons are as follows:

1. Evidence from the first literature which introduces the MASE measure [Hyndman & Koehler, 2006]. The Mean Absolute Scaled Error (MASE) can be calculated by:

$$\text{MASE} = \frac{\frac{1}{n} \sum_{i=1}^n |y_i - \hat{y}_i|}{\frac{1}{n-1} \sum_{j=2}^n |y_j - y_{j-1}|},$$

which scales the MAE error based on one-step naïve forecast computed in-sample, and the denominator of the this metric can be seen as an one-step ahead random walk forecast. While [Hyndman & Koehler, 2006] concluded that MASE greater than one indicate poorer forecasts, this literature particularly emphasized that *"because the scaling is based on one-step forecasts, the scaled errors for multi-step forecasts are typically larger than one"*. And thus, it is reasonable to obtain MASE values greater than one in our experiment. Moreover, they demonstrated that for long-term time series prediction, a smaller MASE score is indicative of better performance, rather than whether the MASE score is smaller than one, as shown in Table 4 of their paper. The evidence presented in this literature (also the first literature to introduce MASE metric) strongly supports our contention that the MASE value of less than one as a criterion for evaluating long-term time series prediction models is inappropriate.

Table 4
Mean absolute scaled error for the M3-forecasting competition

	Yearly	Quarterly	Monthly	Other	All
Theta	2.81	1.97	2.08	1.95	2.20
Theta-sm	2.81	2.00	2.09	1.95	2.22
Robust-Trend	2.63	2.15	2.14	1.92	2.23
Comb SHD	2.88	2.05	2.12	2.09	2.26
ForecastX	2.77	2.22	2.20	1.97	2.31
ForecastPro	3.03	2.35	2.04	1.96	2.33
Dampen	3.03	2.10	2.18	2.08	2.34
HKSG	3.06	2.29	2.15	1.86	2.36
RBF	2.72	2.19	2.27	2.70	2.37
BJ automatic	3.16	2.21	2.21	2.30	2.42
Flores/Pearce1	2.94	2.23	2.31	2.27	2.42
Holt	3.18	2.40	2.15	2.04	2.43
ARARMA	3.48	2.29	2.07	2.05	2.43
SmartFcs	3.00	2.39	2.23	2.08	2.43
PP-autocast	3.02	2.12	2.44	2.09	2.46
Pegels	3.58	2.30	2.13	1.85	2.47
Flores/Pearce2	3.02	2.41	2.27	2.34	2.47
Autobox3	3.18	2.45	2.23	2.01	2.47
Automatic ANN	3.06	2.35	2.34	2.13	2.49
Winter	3.18	2.37	2.43	2.04	2.55
SES	3.17	2.27	2.44	3.14	2.59
Autobox1	3.68	2.61	2.20	2.12	2.62
Naïve2	3.17	2.28	2.50	3.13	2.62
Autobox2	2.75	2.20	3.39	1.90	2.87

All methods were participants in the M3-competition except for HKSG which is based on the method of Hyndman et al. (2002).

2. The label leak problem caused by the naïve forecast for long-term time series forecast. The issue of label leak problem arises from the use of naïve forecast in long-term time series prediction. **Although the MASE value is typically greater than one in such cases, it is erroneous to interpret this as a better performance of the naïve forecast over the multi-step forecasting method.** This is because the naïve forecast uses the label data directly in its prediction, thereby causing a label leak problem if it is utilized as a baseline prediction for multi-step ahead long-term forecast. Consequently, utilizing a MASE value of less than 1 as a criterion to evaluate model performance for long-term time series prediction is inappropriate. Instead, comparing the proposed method with other baselines by considering a smaller MASE score as indicative of better performance, experimental results show that our method achieves the state-of-the-art predictive performance.

Therefore, we respectfully hope you can reconsider your evaluation of our proposed method's performance based on the MASE value of less than one alone. We believe that our work provides valuable insights into long-term time series prediction and deserves further consideration. Thank you for your time and attention to our paper.

[1] Hyndman R J, Koehler A B. Another look at measures of forecast accuracy[J]. International journal of forecasting, 2006, 22(4): 679-688.

REVIEWERS' COMMENTS:

Reviewer #1 (Remarks to the Author):

Dear authors,

I understood that my previous comment is not true. I am very sorry. The paper can be accepted now.